# Technical note: A view from space on global flux towers by MODIS and Landsat: The FluxnetEO dataset

Sophia Walther[1,*], Simon Besnard[1,2,*], Jacob Allen Nelson[1], Tarek Sebastian El-Madany[1], Mirco Migliavacca[1,3], Ulrich Weber[1], Nuno Carvalhais[1,4], Sofia Lorena Ermida[5,6], Christian Brümmer[7], Frederik Schrader[7], Anatoly Stanislavovich Prokushkin[8], Alexey Vasilevich Panov[8], and Martin Jung[1]

[1]Max-Planck-Institute for Biogeochemistry, Hans-Knöll-Straße 10, Jena, Germany
[2]South Pole, Digital Innovation, Fred. Roeskestraat 115, Amsterdam, The Netherlands
[3]European Commission, Joint Research Centre, Via Fermi 2749, Ispra (VA), Italy
[4]Departamento de Ciências e Engenharia do Ambiente, DCEA, Faculdade de Ciências e Tecnologia, FCT, Universidade Nova de Lisboa, 2829-516 Caparica, Portugal
[5]Instituto Português do Mar e da Atmosfera, Rua C do Aeroporto 1749-077 Lisbon, Portugal
[6]Instituto Dom Luiz, Faculdade de Ciências da Universidade de Lisboa, Campo Grande Edifício C1, Piso 1, 1749-016 Lisbon, Portugal
[7]Thünen Institute of Climate-Smart Agriculture, Bundesallee 65, Braunschweig, Germany
[8]V.N. Sukachev Institute of Forest of the Siberian Branch of Russian Academy of Sciences – separated department of the KSC SB RAS, Akademgorodok 50/28, Krasnoyarsk, Russia

**Correspondence:** Sophia Walther (sophia.walther@bgc-jena.mpg.de)

**Abstract.** The eddy-covariance technique measures carbon, water, and energy fluxes between the land surface and the atmosphere at hundreds of sites globally. Collections of standardised and homogenised flux estimates such as the LaThuile, Fluxnet2015, National Ecological Observatory Network (NEON), Integrated Carbon Observation System (ICOS), AsiaFlux, AmeriFlux, and Terrestrial Ecosystem Research Network (TERN) / OzFlux data sets are invaluable to study land surface processes and vegetation functioning at the ecosystem scale. Space-borne measurements give complementary information on the state of the land surface in the surroundings of the towers. They aid the interpretation of the fluxes and support the benchmarking of terrestrial biosphere models. However, insufficient quality, frequent and/or long gaps are recurrent problems in applying the remotely sensed data and may considerably affect the scientific conclusions. Here, we describe a standardised procedure to extract, quality filter, and gap-fill Earth observation data from the MODIS instruments and the Landsat satellites. The methods consistently process surface reflectance in individual spectral bands, derived vegetation indices, and land surface temperature. A geometrical correction estimates the magnitude of land surface temperature as if seen from nadir or 40° off-nadir. Finally, we offer the community living data sets of pre-processed Earth observation data, where version 1.0 features the MCD43A4/A2, MxD11A1 MODIS products, and Landsat collection 1 Tier1 and Tier2 products in a radius of 2 km around 338 flux sites. The data sets we provide can widely facilitate the integration of activities in the eddy-covariance, remote sensing, and modelling fields.

# 1 Introduction

The installation and maintenance of instrumental infrastructure at eddy-covariance (EC) sites worldwide require considerable financial and logistical efforts and labour force. The precious data sets of land-atmosphere fluxes, biometeorological data, and environmental conditions allow fundamental insights on ecosystem functioning (Baldocchi, 2008; Baldocchi et al., 2018; Baldocchi, 2020; Besnard et al., 2018; Migliavacca et al., 2021; Nelson et al., 2020). A significant achievement is the central processing, quality control, and open standardised distribution of a large number of the available observational records in data collections such as the LaThuile, Fluxnet2015, ABCflux (amongst others, Papale et al., 2006; Baldocchi, 2008; Pastorello et al., 2020; Virkkala et al., 2021b; Papale, 2020) to which many site teams contribute.

Complementary information from satellites or digital cameras (phenocams, Wingate et al., 2015) aid and refine studies of local land-atmosphere interactions as they relate to ecosystem structure, phenology, and functioning and the state of the land surface (e.g., Migliavacca et al., 2015; Bao et al., 2022). Earth observation (EO) data for varying regional sizes around the sites can represent the actual area that contributes to the flux measurements - partly even more accurately than similar ground-based measurements can (Gamon, 2015) - provided sufficiently high spatial resolution and temporal overlap with the site-level records. Next to local studies, the combination of flux and satellite observations is also a basic ingredient for upscaling exercises of the in-situ fluxes to larger areas or even the globe (Ueyama et al., 2013; Tramontana et al., 2016; Jung et al., 2019, 2020; Joiner et al., 2018; Reitz et al., 2021; Virkkala et al., 2021a; Zeng et al., 2020).

Independent of the nature of the scientific application, the quality control and gap structure of both the EC and the EO data are the groundwork of each analysis. Different criteria help to identify problematic data points with differing levels of strictness depending on the given application. Moffat et al. (2007) and Falge et al. (2001) describe techniques to fill gaps due to missing data points in the EC data. The literature also offers a diverse set of methods to gap-fill EO data that include spatial, temporal, cross-sensor and cross-variable approaches (to name a few, Wang et al., 2012; v. Buttlar et al., 2014; Weiss et al., 2014; Verger et al., 2011, 2013; Kandasamy et al., 2013; Moreno et al., 2014; Moreno-Martínez et al., 2020; Yan and Roy, 2018; Ghafarian Malamiri et al., 2018; Li et al., 2018; Dumitrescu et al., 2020; Bessenbacher et al., 2021). The pre-processing steps are laborious and they are key to the results and interpretation of the analyses.

We propose a set of systematic pre-processing steps for key land surface indicators from EO data: sub-setting global EO data for an area around an EC site, systematic control for good quality retrievals as well as cloud, snow and water effects, and estimating missing data points in a flexible and ecologically meaningful way. For both the quality control and the gap-filling, the approaches aim to be generalisable across all sites without accounting for specific local conditions, yet flexible enough to accurately reproduce phenological behaviour and characteristic features such as disturbances or fast transitions in managed ecosystems. The procedure shall be as simple as possible, computationally efficient, and not resort to additional data sources to facilitate a potential application to EO data at the global scale.

We apply the proposed processing steps to official data products from the Moderate Resolution Imaging Spectroradiometer (MODIS) instruments and the sensors onboard the Landsat satellites. Both MODIS and Landsat have extensive observational coverage with a high temporal overlap with most freely available EC records. Landsat measurements are of particular interest

because they resolve small spatial details in pixels of 30 m size, but at the cost of missing out on short temporal features. The opposite is true for MODIS data products, which partly average over heterogeneous areas in spatially comparatively coarse pixels of several hundred meters. However, MODIS offers daily, partly even sub-daily temporal resolution. We process EO data sets of both surface reflectance, vegetation indices, and land surface temperature (LST) for a limited area around a given flux site.

As missing data points in EO data are an ubiquitous problem, a number of related initiatives also provide access to EO data that underwent certain pre-processing. For example, Robinson et al. (2017) offer 30m Landsat NDVI for all pixels in the CONUS every 16 days between 1984–2019. They removed cloud effects and filled gaps with climatological averages. Moreno-Martínez et al. (2020) controlled Landsat and MODIS surface reflectance for cloud, snow and water effects and fused them to a gap-free and smoothed product. It covers surface reflectance and its uncertainty in six Landsat spectral bands at monthly, 30m resolution for the CONUS and the years 2009–2020. An example product for gap-free MODIS surface reflectance (as well as albedo and BRDF parameters) at approximately 1km resolution is the MCD43GF product (Sun et al., 2017). In this case, the time series of the parameters of the bidirectional reflectance distribution function are temporally and spatially gap-filled for days and pixels with bad inversion quality or cloud and snow influence, and from those gap-free model parameters a global gap-free product of surface reflectance is provided for the MODIS land bands and three broad spectral bands. Finally, a sub-setting tool (ORNL DAAC, 2018) facilitates access to a range of global EO data sets at a large selection of eddy-covariance sites.

FluxnetEO is unique in proposing the completion of all pre-processing steps necessary for scientific analysis at site-level, hence resulting in an analysis ready dataset. The products cover the period 1984–2017 and 2000–2020 for Landsat and MODIS, respectively, and are freely available by the services of the ICOS Carbon Portal (see data availability statement, Walther et al. (2021a, b)). Each data set has a complementary data layer with additional flags to inform the user whether data points correspond to actual good quality observations according to the proposed criteria and, if not, how they have been estimated in different gap-filling steps. FluxnetEO provides a ready-to-use dataset, which, however, means limited flexibility for the users to make their own decisions on the pre-processing steps. For example, they depend on the site selection made by the authors (see table E1 for the site selection in version 1.0) and their decision to cover an area within a radius of 2 km around a site. Conversely, the ORNL DAAC (2018) offers larger cutout radii of 4 km around a considerably larger collection of sites than FluxnetEO and from a complementary selection of global EO products. But users will need to invest considerable work in quality control and gap-filling. Regarding available quality controlled and gap-free large scale or even global gridded EO data (Moreno-Martínez et al., 2020; Robinson et al., 2017; Sun et al., 2017), the user needs to find ways to access these data sets at site level (while Moreno-Martínez et al. (2020) is available on Google Earth Engine (GEE), Sun et al. (2017) is not, Robinson et al. (2017) needs shape files), and needs to understand whether the applied quality filters match the needs of their application.

To allow potential users to make an informed decision on the product which suits their application best we describe details about data inputs in FluxnetEO in section 2.2, explain the quality control and gap-filling approaches in section 3, illustrate examples and benchmark the products against a selection of independent products and approaches in section 4. Table 2 and the data availability section provide detailed information on the resulting products, while table 1 summarises and compares the main characteristics of the selected studies and services mentioned above (Robinson et al., 2017; Sun et al., 2017; Moreno-Martínez et al., 2020; ORNL DAAC, 2018) and the one in this contribution. We expect FluxnetEO to be a living data set with regular updates regarding the site selection, the temporal coverage, the release of new Landsat/ MODIS collections and processing improvements based on user feedback. Potential users are therefore advised to refer to the ICOS Carbon Portal for the latest product version and site availability information (Walther et al., 2021a, b)).

## 2 Data

### 2.1 Eddy-covariance sites

For the current version 1.0 of the product we select the 338 sites from the LaThuile, Fluxnet2015 (Pastorello et al., 2020) and ICOS Drought 2018 Initiative (Drought 2018 Team and ICOS Ecosystem Thematic Centre, 2020) flux data releases. Site coordinates given in different sources (Ameriflux, Asiaflux, Europe-Fluxdata, Fluxdata.org, and a previously compiled in-house Fluxnet-site location list) may differ. In that case, the coordinates with the highest precision were selected. In case the coordinates differed by more than 0.001° for a given site, a manual check in Google Earth identified the correct or most probable location of the site. The final set of 338 sites for which we process the MODIS and Landsat EO data in product version 1.0 is listed in table E1. Forests and grasslands are best represented among the 338 sites. The collection includes fewer sites from savannas and shrublands, and only one site from a deciduous needleleaf forest (table 1).

### 2.2 MODIS and Landsat

The MCD43A4 product combines AQUA and TERRA observations and provides estimates of surface reflectance in the MODIS bands 1-7 (Schaaf and Wang, 2015b). Time series represent observations modelled at nadir view at a resolution of 16 days and 500 m spatial pixels. For the quality control of MCD43A4, a complementary product, MCD43A2, contains band specific information on the quality of the inversion of the bidirectional reflectance distribution function as well as snow cover, platform information and land/water coverage in the scene (Schaaf and Wang, 2015a).

The MODIS MOD11A1 (TERRA, starting in 2000) and MYD11A1 (AQUA, starting in 2002) products (hereafter jointly referred to as MxD11A1, Wan et al. (2015a, b)) provide daily LST and emissivity estimates aligned with quality and view angle information at 1 km spatial pixel sizes. The LST values represent instantaneous values and are selected based on viewing zenith angle and LST values (MOD11A1 user guide, https://lpdaac.usgs.gov/documents/118/MOD11_User_Guide_V6.pdf).

**Table 1.** Representation of different plant functional types and Koeppen climate classes across the 338 sites in the FluxnetEO v1.0 collection.

| plant functional type | number of sites | Koeppen main climate | number of sites |
|---|---|---|---|
| evergreen needleleaf forest (ENF) | 86 | arid | 26 |
| evergreen broadleaf forest (EBF) | 25 | equatorial | 23 |
| deciduous needleleaf forest (DNF) | 1 | warm temperate | 171 |
| deciduous broadleaf forest (DBF) | 40 | snow | 103 |
| mixed forest (MF) | 13 | polar | 12 |
| woody savanna (WSA) | 10 | undefined | 3 |
| savanna (SAV) | 11 | | |
| closed shrubland (CSH) | 6 | | |
| open shrubland (OSH) | 19 | | |
| grassland (GRA) | 58 | | |
| crops (CRO) | 36 | | |
| wetlands (WET) | 32 | | |
| snow (SNO) | 1 | | |

Four LST data streams are available: TERRA$_{day}$ with observations around 10.30 am local time, AQUA$_{day}$ with observations around 1.30 pm, TERRA$_{night}$ around 10.30 pm and AQUA$_{night}$ around 1.30 am. For each of them, observation times vary between overpasses by about $\pm 1.5$ hours.

Observation geometries need special attention as the MODIS instruments measure in a wide swath to obtain high temporal
coverage. They scan across their track from right to left with view zenith angles up to 65 degree from nadir. The wide range of viewing geometries leads to different fractions of surface types seen from one overpass to the next for a given site. In addition, vegetation structure and topography, together with the position of the sun relative to the sensors, cause variable shadowing effects. The reflectance product (MODIS MCD43A4, Schaaf and Wang (2015b)) partly accounts for these anisotropy effects and simulates a nadir view. In order to partly account for variability in the observed LST that is related to changing observation
geometry (Rasmussen et al., 2011; Guillevic et al., 2013; Ermida et al., 2014), a correction approach developed by Ermida et al. (2018) estimates an LST offset as if the instrument would measure from directly above a site. For some applications, an oblique view might be favourable over a nadir constellation, for example to enhance the contribution of vegetation canopy to the LST estimate and minimise fractions of soil or understorey. In addition, we provide LST corrected to a viewing zenith angle of 40 degrees.

Reflectance-based Landsat time series comprise the entire multi-temporal collection 1 of the Landsat 4, 5, 7 and 8 archives (https://landsat.gsfc.nasa.gov/data) covering the period 1984-2017 at 30 m spatial pixel size. The seven spectral bands of the Landsat product were collected: BLUE, GREEN, RED, near infrared (NIR), shortwave infrared 1 and 2 (SWIR1, SWIR2),

and thermal infrared (TIR) (https://landsat.usgs.gov/what-are-band-designations-landsat-satellites). Landsat data have been pre-processed using the Landsat Ecosystem Disturbance Adaptive Processing System (LEDAPS, Schmidt et al., 2013) and the Landsat Surface Reflectance Code (LaSRC, https://landsat.usgs.gov/landsat-surface-reflectance-data-products) for atmospheric correction. The pixelQA layer contains information related to clouds, cloud shadows, snow, and ice and is useful for the quality control of the Landsat data (Zhu and Woodcock, 2012; Zhu et al., 2015). In contrast to MODIS, the Landsat sensors acquire images at much smaller view angles around 7.5-degree from nadir. Ground control points and a digital elevation model help to correct for small directional effects related to terrain structure and viewing angles (Wulder et al., 2019). Corrections for the small but significant differences between the spectral characteristics of Landsat ETM+ and OLI (Roy et al., 2016) are not applied.

The services by GEE provided cutouts of the above mentioned products at the EC sites. Independently of the product and its spatial resolution, the cutout area was limited to a maximum distance of 2 km between a given tower and the centre of a given satellite pixel. No single cutout size will fit the flux footprint extents of all sites (Chu et al., 2021). The decision for a radius of 2 km in product version 1.0 compromises reasonable data set sizes and the inclusion of the high temporal resolution flux footprints for the majority of sites. Downloading the EO data in tiff-format avoided intransparent re-projection of the data from sinusoidal to regular grid by GEE, which would have been problematic for the quality flags in the MCD43A2 and MxD11A1 products. The Landsat data were already provided in regular grid by GEE.

## 3 Methods

We describe here the overall concept and rationale of the quality filter and the gap-filling, but report all technical details in the Appendix A.

### 3.1 Processing steps of reflectance-based indicators

The processing steps for reflectance-based land surface variables can be summarised by the following steps:

1. quality control for effects of snow, water, bad inversion per spectral band and individual pixel in a cutout (henceforth subpixel) using the MODIS/Landsat quality flags

2. optionally compute vegetation indexper subpixel, or use the raw spectral bands

3. optionally spatially aggregate over a selection of subpixels in the cutout to obtain one time series per site, or decide to process all subpixels individually

4. remove values of an index outside its defined ranges and apply an additional outlier filter

5. gap-filling

### 3.1.1 Quality control and computation of spectral indices

Quality control of the MODIS reflectance-based vegetation indices focused on three aspects: good inversion quality of the bidirectional reflectance distribution function as indicated by the BRDF_Albedo_Band_Quality_Bandx flags in the MCD43A2 product, snow-free conditions according to the Snow_BRDF_Albedo flag, and the omission of reflectance values that are affected by the presence of water in the field of view using the BRDF_Albedo_LandWaterType flag. For the selected data samples which passed those criteria we computed a large set of spectral vegetation indices (table 2). An additional check removed possible values of the vegetation indices outside their defined ranges. Some of the time series contained obvious outlier values. We employed an empirical filter which largely removed those samples which had a particularly large difference to the median of their surrounding values in a temporal window (Papale et al., 2006, technical details on all filters in Appendix A).

In the Landsat data, the flag pixel_qa provided quality attributes (CFMask, Foga et al., 2017) and removed pixels that contained snow/ice, water, cloud, and/or cloud shadow using a binary flag of presence. Similar to the MODIS product, we computed a series of spectral vegetation indices (table 2) using the good quality observations and removed possible values of the indices outside their defined ranges. A slightly modified filter removed possible outlier values also for the Landsat data (see details in Appendix A.)

### 3.1.2 Gap-filling

In the literature several gap-filling and smoothing approaches are available which work in one or more dimensions (e.g., Wang et al., 2012; Kandasamy et al., 2013; v. Buttlar et al., 2014; Weiss et al., 2014; Yan and Roy, 2018; Zhang et al., 2021) or use fusion methods between sensors (Verger et al., 2011; Moreno-Martínez et al., 2020). They differ in their levels of sophistication and computational efforts. One of our requirements for the gap-filling approach was that it employs exclusively temporal operations and does not use additional data sources. It is hence very generalisable and allows the gap-filling to be generally applicable to a single time series per site, several subpixels in a cutout around a site, and global EO data. A number of possible applications will require the analysis of actual observations, and consequently approaches that fit smooth functions to available good quality data (e.g., Jonsson and Eklundh, 2002; Gonsamo et al., 2013) to represent a gap-free time series are not suitable. Therefore, the idea was to retain the good quality data and make as realistic estimates as possible for the gaps between them. The following recipe describes the steps to estimate missing data points conceptually, all technical details we report in Appendix A. Unless stated otherwise, for each gap-filling step, the values filled in previous steps guide the current and subsequent gap-filling steps together with the good-quality observations.

1. Fill short non-snow related gaps ($\leq 5$ days or $\leq 1$ month for MODIS and Landsat, respectively) with a median across valid values in moving windows of 16 days (3 months for Landsat). The moving median only fills gaps, it does not change/ smooth valid data points.

2. Fill snow related gaps with a constant baseline value which is identified as the average of valid data points adjacent to snow covered periods, i.e. immediately before snow fall or after snow melt (after Beck et al., 2007, but see details in

Appendix A). Consider all times with a snow flag larger than 0.1 or missing snow information as snow covered. The latter periods are included as the snow flag appears to systematically miss snow periods in higher latitudes in the beginning of the winter. Still, frequent gaps with missing snow information also occur during the growing season. In order to avoid wrong filling with a constant value during the growing season this gap-fill step is not applied when the probability of snow cover is low, i.e. when the average seasonal cycle indicates typically snow-free conditions at a given time of the year, or when typically no snow occurs at all at a given site.

3. Subsequently, another moving median in windows of 40 days (4 months for Landsat) fills gaps shorter than 65 days (2 months for Landsat).

4. Linearly regress the time series on its own median seasonal cycle (MSC). Compute a re-scaled MSC with the obtained regression parameters and use it to fill longer gaps. Execute the regression and re-scaling in temporal moving windows as this guarantees more flexibility to correctly represent inter-annual variations in the time series and even partly accounts for changes in the shape of the seasonal cycle due to disturbances. It is, however, not suited to fill regularly recurring gaps at a certain time of the year, e.g. during rain seasons (Verger et al., 2013).

5. Fill the remaining gaps by piecewise cubic polynomial interpolation. Time series with less than 300 valid data points in the whole record after application of all the previous gap-filling steps will not be meaningful for analysis but are still filled by nearest neighbour interpolation.

6. Temporal operations cannot meaningfully fill gaps at the beginning and at the end of the record. Therefore the first/last valid data points are repeatedly appended at the beginning/end of the record.

The described processing steps are generalisable across a range of spectral vegetation indices and can reliably fill missing data points across sites globally (see examples in section 4). However, a number of sites have extremely low data availability after quality checks, and the gaps in their time series are challenging to temporally interpolate in a meaningful way. This can lead to problematic gap-filled data points with questionable reliability and realism. Examples are tropical sites and/ or sites with a pronounced wet season with permanent cloud cover. The same generally applies for MODIS in the years 2000-2002 when observations stem mainly from the TERRA satellite and therefore data availability is comparatively low. For Landsat, the number of available scenes is relatively heterogeneous across the globe (https://www.usgs.gov/media/images/cumulative-number-scenes-landsat-archive) with some regions having a very good coverage (e.g., North America) while other regions are observed less frequently (e.g., Russia and Africa). Such differences in the availability of good quality data between sites strongly affect the quality of the gap-filling at site level. In addition, FluxnetEO provides for each data layer a gap-fill flag, consistsing of a range of integer values to identify original good quality data (flag=0) from gap-filled estimates (flags=1...n) where information is provided in which gap-filling step a certain data sample has been imputed. This allows users to explore individual sites and use (parts of) the gap-filled data or resort to only using the high quality original data points.

## 3.2 Preprocessing of MODIS land surface temperature

The processing of the LST follows this order:

- outlier filter for each LST data stream and check that any daytime LST is higher than any nighttime LST per subpixel

- optionally apply a geometrical correction per subpixel

- optionally aggregate over a selection of subpixels in the cutout per time step and LST data stream

- gap-fill the aggregated time series or each subpixel for all four MODIS LSTs simultaneously

### 3.2.1 Quality checks

The quality control of the MODIS LST focused on removing outlier values. Negative outlier values in LST might represent residual cloud contamination, whereas unusually high values might originate from undetected saturation in the level 1 data. We found that the flags provided in the MxD11A1 products are insufficient to achieve this. Instead, empirical quality checks followed the procedure for the MODIS reflectances, i.e. they discarded data points that deviated strongly from the median of their surrounding values in temporal windows of 30 days (Papale et al., 2006). An additional sanity check eliminated any daytime LST lower than the minimum of AQUA and TERRA nighttime LST for a given day.

### 3.2.2 Geometrical correction

For several applications, variable viewing geometries as inherent in the MODIS LST observations are not desirable. A geometrical correction approach developed by Ermida et al. (2018) accounted for directionality in LST retrievals due to vegetation structure and topographical effects. A parametric model estimates the magnitude of LST as if constantly observed from nadir or an angle of 40 degrees between the sensor and the zenith above a given site. Ermida et al. (2018) derived the coefficients for this geometrical model at a resolution of 0.05 degree. We followed the pragmatic approach of selecting the model coefficients for the correction from the pixel containing a given site. We acknowledge that we did not investigate to what extent the given site conditions represent the overall characteristics of the land surface in the allocated pixel. Further input to the geometrical model were the viewing azimuth angles, solar angles at the overpass time and estimates of daily potential radiation at the top of the atmosphere. The geometrical correction was applied to each subpixel in a cutout separately.

### 3.2.3 Gap-filling

Also for the gap-filling of LST several approaches are present in the literature (e.g., Gerber et al., 2018; Ghafarian Malamiri et al., 2018; Li et al., 2018; Dumitrescu et al., 2020). When using exclusively operations in time and no ancillary data to estimate invalid LST observations, one needs to consider the shorter autocorrelation of LST compared to the reflectance-based indicators. According to Vinnikov et al. (2008), the weather-related component of clear-sky LST has an autocorrelation of about 3 days. The following sequence of steps filled the four MODIS LST data streams (for technical details refer to Appendix B):

1. Similar to the reflectances, a first step consisted of a temporal moving median in windows of eight days to fill gaps.

2. A second step was inspired by Li et al. (2018) and Crosson et al. (2012) and foresaw to use one of the four MODIS LST time series as a 'reference' to fill gaps in a second 'imputed' one. We computed a MSC of the difference between the 'reference' and the 'imputed' MODIS LST. This average shift was linearly scaled to the actual shift in temporal windows. The scaled average shift added to the 'reference' LST represented the values used to fill gaps in the 'imputed' LST time series. This procedure iteratively used three of the MODIS LST data streams to fill the fourth, i.e. each one is imputed once by all three others (see details in Appendix B). This gap-fill step was only possible in cases where not all four MODIS LST observations were invalid during a given day, but extremely advantageous to preserve short synoptic variability in the gap-fill estimates.

3. In fully cloudy days without any valid LST observation, or in case a period has too few valid observations for a meaning-ful calibration of the linear model in the previous step. The gap-filling followed the same steps as for the reflectance-based spectral indices:
   In temporal windows, find a linear scaling between one LST time series and its own MSC. Use the slope and intercept parameters to compute a re-scaled MSC, which fills gaps in the time series for days of the year when the MSC is valid.

4. Interpolate the remaining gaps with cubic polynomials, or nearest neighbour in case of very low data availability (less than 300 valid data points in the entire time series).

5. Missing values at the beginning and the end of the record cannot be meaningfully filled by temporal methods and are therefore simply repeated.

Steps 3-5 produced very smooth and, therefore, less realistic LST estimates than steps 1-2. Also, one needs to be aware that any LST estimate in data gaps from this procedure necessarily represents an LST estimate under clear sky conditions, which can be very different from the real LST under overcast skies (Ermida et al., 2019). This needs to be considered for a given application to prevent the effects of clear-sky bias in the LST data sets on the results. Like the vegetation indices, LST data layers have a gap-fill flag in FluxnetEO describing which data points are original and which gap-filling step filled the missing values.

## 3.3 Evaluation and benchmarking

### 3.3.1 FluxnetEO performance in comparison to a machine learning approach (missForest)

A common approach to benchmarking gap-filling methods is to artificially remove samples at positions where the true data value is known, then subject the time series to the gap-filling approach and compare the gap-filled estimates with the original values (Moreno-Martínez et al., 2020; Zhang et al., 2021; v. Buttlar et al., 2014; Wang et al., 2012; Verger et al., 2011, 2013; Gerber et al., 2018). We apply this approach to FluxnetEO in artificial gaps for MODIS and Landsat variables, and randomly

remove 20% and 40% of data samples (corresponding to a low and medium gap fraction, compare Fig.1) per site at positions with originally good quality. We remove data points from a gap-free time series, i.e. the data points which had been gap-filled before guide the gap-filling in the artificial gaps. We feed the time series of the station pixel with artificial gaps into the gap-filling approaches described in section 3 and quantify the gap-filling performance compared to the true values with the Nash-Sutcliffe efficiency  (NSE, Nash and Sutcliffe, 1970). NSE close to one indicates good performance, while negative values mean worse performance than inputting the simple average into the gaps. Decidedly, the NSE refers exclusively to the data samples from the artificial gaps and not to the complete time series.

To have an independent benchmark of FluxnetEO, we compare to the performance of a versatile imputation method, missForests (Stekhoven and Bühlmann, 2011), in the same artificial gaps. MissForest is based on random forests and can handle variables of different types and dimensions. It is a multi-output machine learning method that iteratively fills gaps across variables, considering their potential non-linear dependencies. We input all MODIS (Landsat) variables per site together with the information on snow fraction and the day of year or month of year for MODIS or Landsat, respectively. Hence, per site and mission, missForest iteratively imputes all variables collectively.

### 3.3.2   Comparison with other gap-filled data sets: Moreno-Martínez et al. (2020)

A complementary and mandatory approach to assessing the quality and characteristics of the proposed pre-processing steps is a comparison against independent data sets and approaches (e.g.  Moreno-Martínez et al., 2020; Robinson et al., 2017; Sun et al., 2017). Different spatio-temporal resolutions in the provided data sets and the fact that often mass downloads of data are necessary to evaluate them at site-level challenge this approach. However, Moreno-Martínez et al. (2020) provide their gap-filled Landsat surface reflectance at the same spatio-temporal resolution like FluxnetEO, and access and cutout at the site-level via GEE is feasible. We, therefore, compare the FluxnetEO Landsat product and the  Moreno-Martínez et al. (2020) surface at 86 sites in the CONUS for the years 2009-2017, which corresponds to the spatiotemporal domain in which both are available. In the comparison, we do not differentiate between original good quality and gap-filled estimates because quality control and, therefore, gap-structure differ between the products. However, unphysical reflectance values lower than 0 or larger than 1 occur, especially in winter, and were removed before the cross-consistency analysis, both from good quality and gap-filled estimates.

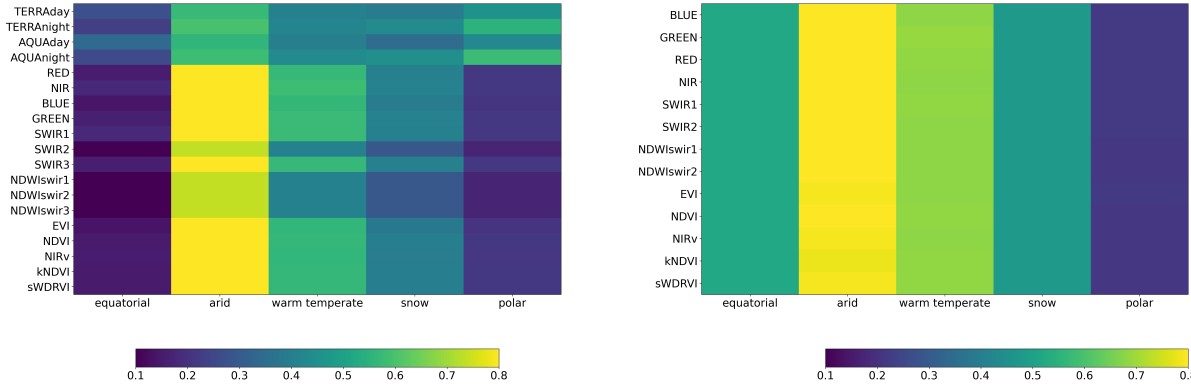

**Figure 1. Fraction of good quality data in the MODIS (left) and Landsat (right) time series.** Colours represent the median data availability in tower pixels across sites grouped by Koeppen climate classification. Data refer to the period 2003–2020 for MODIS (the time period when both TERRA and AQUA satellites are in space) and 1990–2017 for Landsat.

## 4  Results and Discussion

### 4.1  Gap-statistics across indices

Data availability after quality screening is highly variable between sites and depends on the data stream (Fig. 1). Large differences in the amount of good quality data in groups of different climate regions, especially for the reflectances, mirror general atmospheric conditions in different regions. Differences between spectral bands and reflectance-based indices are very minor in both MODIS and Landsat. MODIS LST generally has less valid data points among the data sets than the reflectance-based indicators, and often less during daytime than nighttime. While the LST are instantaneous values, the reflectances represent averages over 16-day periods. A lower number of good quality observations in indices that rely on band 6 relate to degraded detectors in AQUA MODIS band 6.

### 4.2  Temporal patterns of the gap-filled time series

We illustrate some characteristics of the time series in FluxnetEO using the pixel containing an EC station at example sites. The Austrian site Neustift (AT-Neu) was situated in a valley in the Alps and surrounded by grasslands which were typically mown three times a year (Wohlfahrt et al., 2008). According to their nature, the MODIS LST time series exhibit faster variability than the vegetation indices (Fig. 2). Midday observations (AQUAday) partly show an LST increase after the first harvest event in a year around the day of the year 150. The MSC of most vegetation indices clearly marks the mowing timing, although the relative magnitude varies between indices. Constant values in winter represent snow-covered times. For Landsat, the granularity

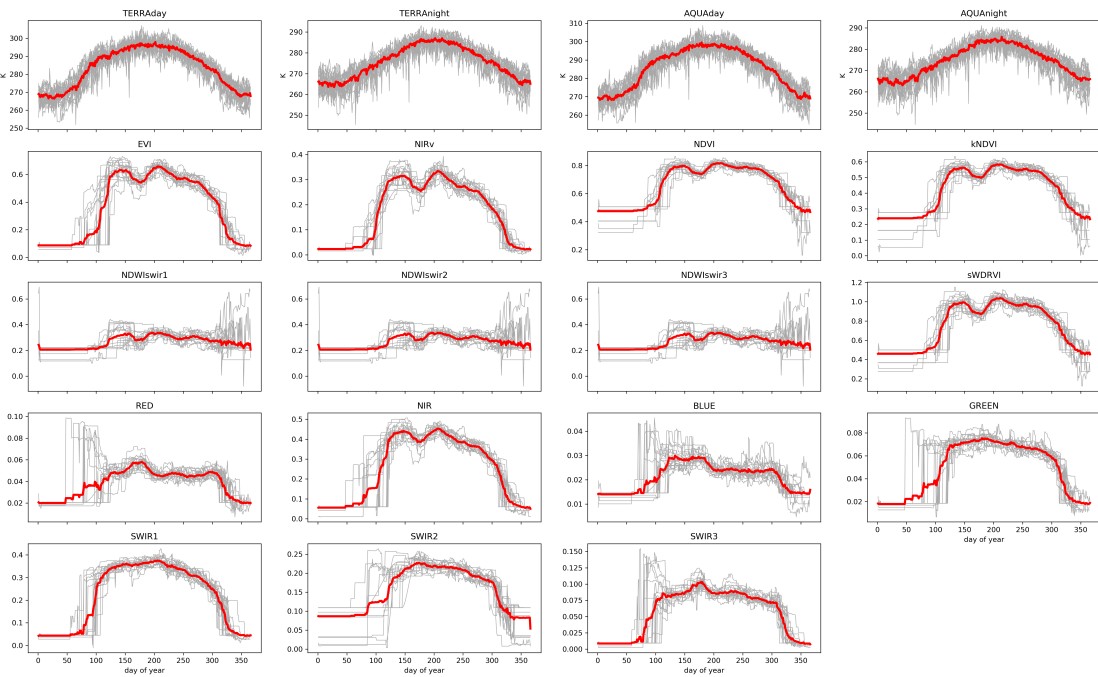

**Figure 2.** Median seasonal cycle (red) and individual yearly trajectories (gray) for MODIS LST (top row) and MODIS vegetation indices and surface reflectance (second to last rows) in the pixel containing the Austrian site Neustift (AT-Neu). Depending on the data set the central pixel measures 500m or 1km.

of temporal patterns is clearly lower due to the monthly sampling, but the characteristic management effects are visible also here (Fig. 3).

Focusing on the example of the EVI, other sites illustrate a few characteristics of the gap-filling procedure in more detail
(Fig. 4, 5): At the evergreen needleleaf forest site El Saler in Spain (ES-ES1) much data passes the quality control and mostly short gaps are reliably filled also in the absence of a very regular seasonal cycle in EVI in both MODIS and Landsat. The boreal forest site Saskatchewan (CA-SF1) illustrates the effect of a disturbance that happened in 2015 (though the site was operated only until 2006). The gap-filling procedure adapts to the modified conditions both abruptly when the disturbance happens and gradually during recovery in the following years. There is a problematic group of high MODIS EVI values during winter
2006/07. The moving window outlier filter applied to the MODIS reflectances is by design unable to detect those outliers as they occur consecutively in a short period of time. Tharandt (DE-Tha, evergreen needleleaf forest) and Lonzee (BE-Lon, crops), are examples of the challenges that data scarce periods bring for both Landsat and MODIS. For MODIS, estimated values in the years 2000-2002 (where only TERRA was in operation) are less reliable at both sites. Landsat is particularly scarse

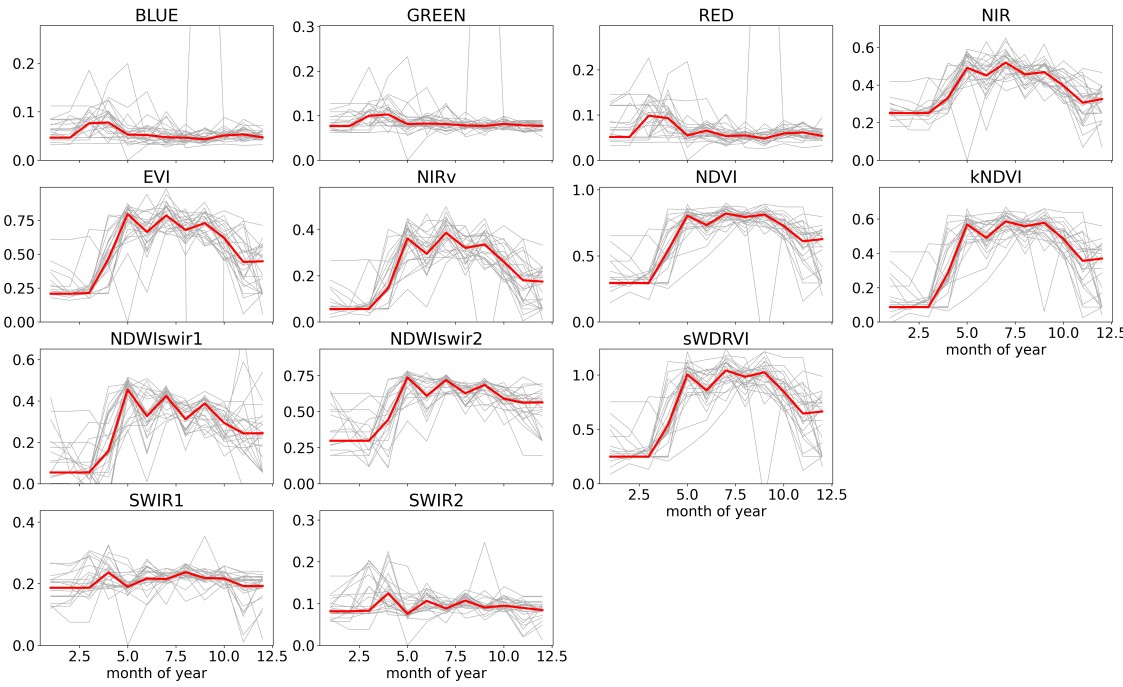

**Figure 3.** Median seasonal cycle (red) and individual yearly trajectories (gray) of the different data sets in the 30m pixel containing the Austrian site Neustift (AT-Neu) Landsat.

and the gap-filling unsuccessful at Tharandt in the eighties, in 1994/95 and 2008-12, and in Lonzee a clear seasonality in EVI
establishes only after 2000. In addition, for MODIS false filling by the snow baseline value during the growing season could
not entirely be prevented, causing an unrealistic dip in one year in each of the sites. Note that the snow flag contains partly long
data gaps in CA-SF1, DE-Tha and BE-Lon. Finally, the woody savanna site Adelaide River (AU-Ade) is a typical example of
EC sites in climates with a dry and a wet season. While in the dry season basically no data gaps occur, cloud coverage in the
rainy season is long enough such that mainly the last gap-filling steps of a linearly scaled MSC and interpolation take effect for
MODIS (Fig. 2). Although the scaling of the MSC does not fully succeed in all years to produce smooth transitions between
the good quality data and the gap-filled ones, the interpolation is able to preserve inter-annual variations in the MODIS EVI.

Missing MODIS LST values were estimated most reliably in the gap-filling steps 1-2 (moving median and scaled average
shift to observations at other overpass times) because the typical short-term variability in the time series could be preserved.
In the Spanish site Majadas de Tietar (ES-LMa, Fig. 6 top panel), savanna-type vegetation is prevalent with a dry summer and
wet winter. Visually the gap-filling procedure succeeds in preserving the typical higher LST variability in the dry season and

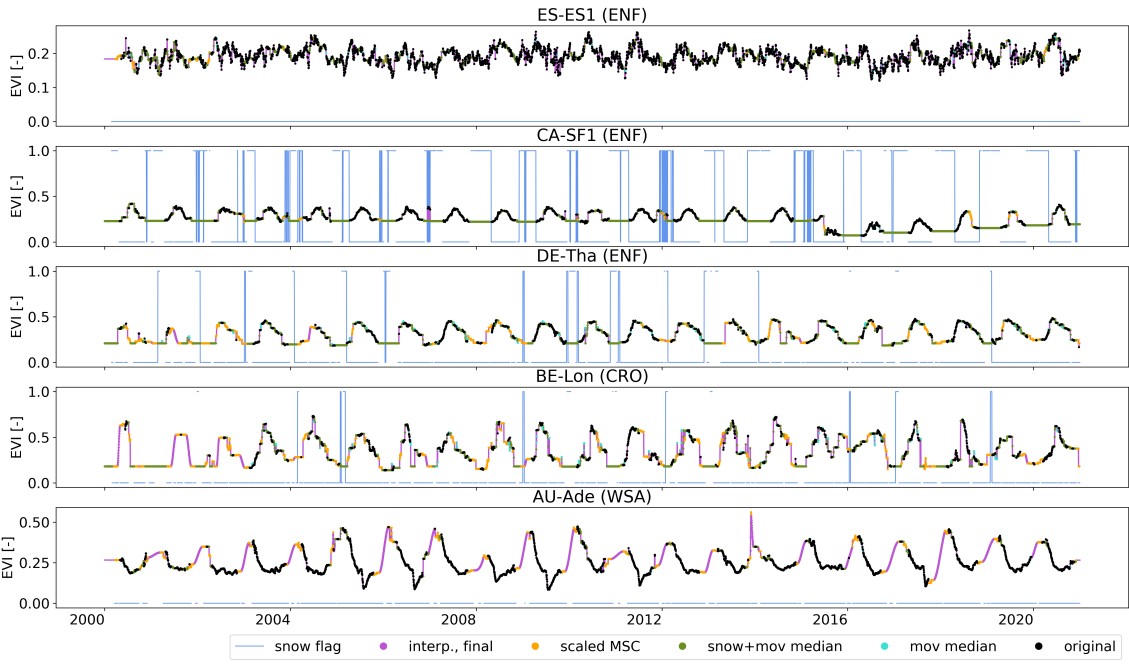

**Figure 4.** Illustration of gap-filling steps in the 500m pixel containing selected eddy-covariance sites for the MODIS EVI.

seasonally changing diurnal amplitudes. Also, in Saskatchewan (CA-SF1), gap-filling step 2 successfully estimates the largest fraction of missing values for each data stream from the complementary observation times. The EVI indicated a disturbance event at the beginning of 2015 (Fig. 4) that continued to strongly affect the EVI also in the following year. The event also marks the LST time series in that daytime LST, and therefore, the diurnal amplitude clearly increases in summer after 2015. The gap-filling procedure follows this behaviour. Relative to Majadas de Tietar or Saskatchewan, in the mixed forest in Vielsalm (BE-Vie), data gaps are much more persistent throughout a day and the gap-filling works more often with the third gap-filling step using an average seasonal cycle of LST to estimate missing observations. Finally, at the woody savanna site Howard Springs in northern Australia (AU-How, Fig. 6 bottom panel) there is a strong seasonal phasing between daytime and nighttime LST. Data availability also changes with the seasons. In the monsoon season, synoptic variability in the filled data points is unrealistically low because the gap-filling needs to resort to filling by a median seasonal cycle of LST (obtained from those years in which the monsoon starts late) or by interpolation.

Geometrical corrections to the nadir viewing angle are much larger and have a stronger seasonality for daytime LST than for nighttime observations (rightmost panel in Fig. 6, Ermida et al. (2018)). The daytime LST value from a nadir view is consistently estimated to be several Kelvin higher than from an oblique view. The Australian Howard Springs is an exception in that the correction offset to nadir has no consistent sign during the wet season.

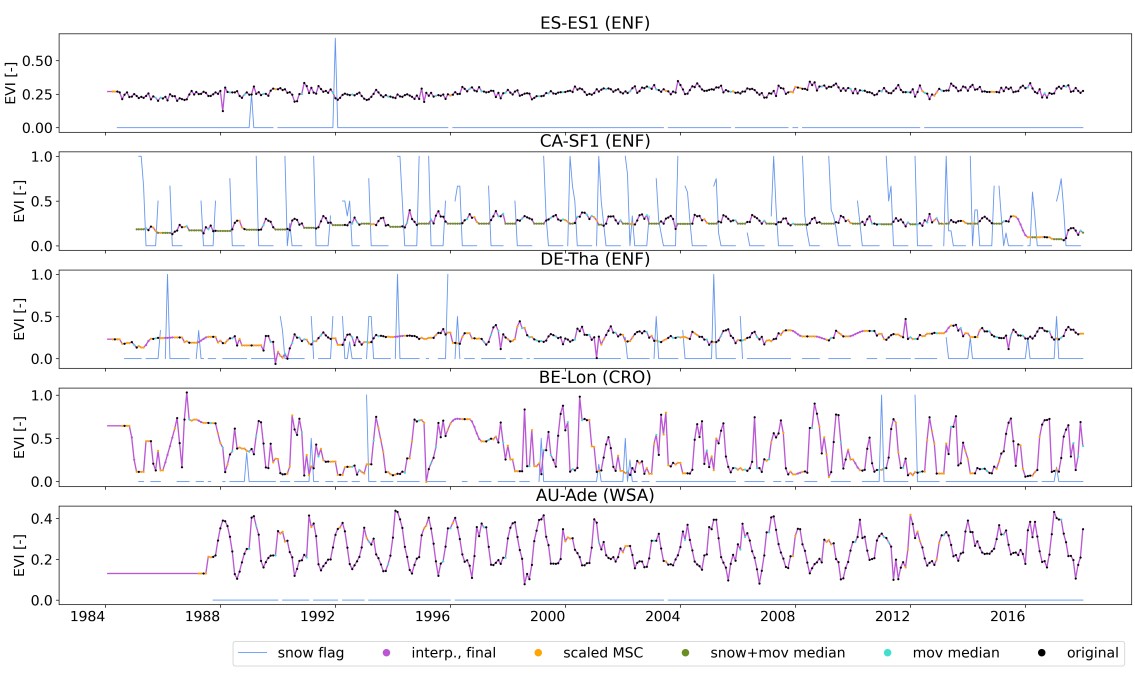

**Figure 5.** Illustration of gap-filling steps in the 30m pixel containing selected eddy-covariance sites for the Landsat EVI.

Table 2. **Data sets presented in FluxnetEO.** b*x* refer to the spectral bands. Each data set spans the time period 2000-2020 (MODIS) and 1984-2017 (Landsat) and contains a flag describing whether a data point is good quality or whether and how it has been estimated in the gap-filling procedures.

| index/ variable | MODIS | Landsat 4,5 and 7 | Landsat 8 | notes |
|---|---|---|---|---|
| *reflectance-based indicators* | | | | |
| EVI | $2.5 \cdot \frac{b2-b1}{b2+6*b1-7.5*b3+1}$ | $2.5 \cdot \frac{b4-b3}{b4+6*b3-7.5*b1+1}$ | $2.5 \cdot \frac{b5-b4}{b5+6*b4-7.5*b2+1}$ | Huete et al. (2002) |
| NDVI | $\frac{b2-b1}{b2+b1}$ | $\frac{b4-b3}{b4+b3}$ | $\frac{b5-b4}{b5+b4}$ | Tucker (1979) |
| kNDVI | $tanh\left(\frac{b2-b1}{b2+b1} \cdot \frac{b2-b1}{b2+b1}\right)$ | $tanh\left(\frac{b4-b3}{b4+b3} \cdot \frac{b4-b3}{b4+b3}\right)$ | $tanh\left(\frac{b5-b4}{b5+b4} \cdot \frac{b5-b4}{b5+b4}\right)$ | Camps-Valls et al. (2021) |
| NDWIswir1 | $\frac{b2-b5}{b2+b5}$ | $\frac{b4-b5}{b4+b5}$ | $\frac{b5-b6}{b5+b6}$ | Gao (1996) |
| NDWIswir2 | $\frac{b2-b6}{b2+b6}$ | $\frac{b4-b7}{b4+b7}$ | $\frac{b5-b7}{b5+b7}$ | Gao (1996) |
| NDWIswir3 | $\frac{b2-b7}{b2+b7}$ | - | - | Gao (1996) |
| NIRv | $(NDVI - 0.08) \cdot b2$ | $(NDVI - 0.08) \cdot b4$ | $(NDVI - 0.08) \cdot b4$ | Badgley et al. (2017) |
| sWDRVI | $\frac{(0.3-1)+(0.3+1)*\frac{b2-b1}{b2+b1}}{(0.3+1)+(0.3-1)\cdot\frac{b2-b1}{b2+b1}} + \frac{1-0.3}{1+0.3}$ | | | Gitelson (2004) |
| RED | b1 | b3 (RED) | b4 (RED) | |
| NIR | b2 | b4 (NIR) | b5 (NIR) | |
| BLUE | b3 | b1 (BLUE) | b2 (BLUE) | |
| GREEN | b4 | b2 (GREEN) | b3 (GREEN) | |
| SWIR1 | b5 | b5 (SWIR1) | - | |
| SWIR2 | b6 | - | b6 (SWIR1) | |
| SWIR3 | b7 | b7 (SWIR2) | b7 (SWIR2) | |
| *MODIS land surface temperature* | | | | |
| LST | TERRA$_{day}$, TERRA$_{night}$, AQUA$_{day}$, AQUA$_{night}$ | | | each with variable viewing zenith angle |
| LST_nadir | TERRA$_{day}$, TERRA$_{night}$, AQUA$_{day}$, AQUA$_{night}$ | | | corrected to viewing zenith angle = 0 degrees |
| LST_oblique | TERRA$_{day}$, TERRA$_{night}$, AQUA$_{day}$, AQUA$_{night}$ | | | corrected to viewing zenith angle = 40 degrees |

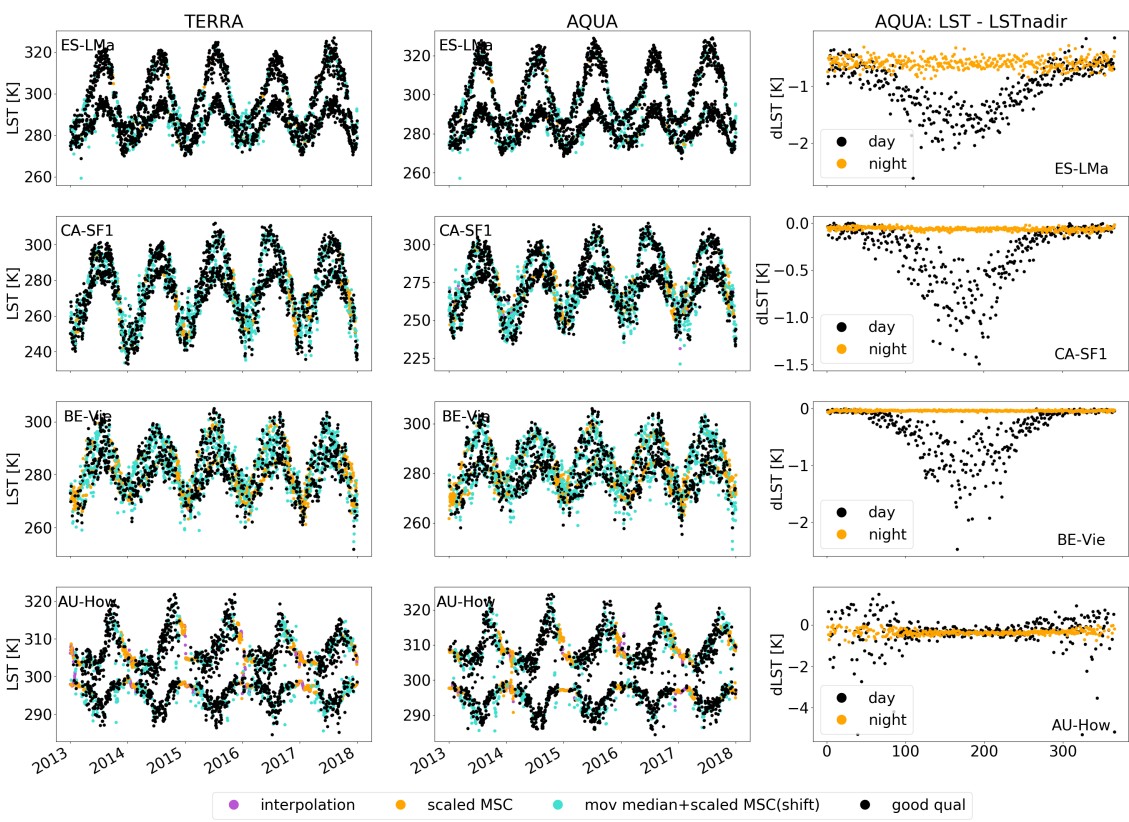

**Figure 6.** MODIS LST gap-filling steps in the 1km pixel containing selected eddy-covariance sites for daytime and nighttime LST. The rightmost column shows the average annual cycle of the correction factor between LST from variable viewing angles and LST corrected to nadir view.

## 4.3 Benchmarking

In the experiments where artificial gaps are introduced at data points with known and valid values in the pixel containing the eddy-covariance site, FluxnetEO performance for MODIS is excellent with NSE values clearly above 0.9 for all reflectance-based indices, and even above 0.95 for artificial gap fractions of 20% (Fig. C1 top left). The NSE of the gap-fill estimates for LST is systematically lower, but above 0.8 and therefore still very good. Interestingly, the median NSE across sites are very similar for the 20% and 40% gap fraction experiments for the LST, but clearly different for the reflectance. Overall, FluxnetEO outperforms missForest in the realism of the gap-fill estimates slightly but consistently across most reflectance-based MODIS variables, and more strongly so for the larger (and more realistic for the majority of sites) artificial gap-fraction of 40% (Fig. 7 left). The NDWI variables are a special case, where missForest does not succeed to produce reliable estimates (Fig. C1 top right), and interestingly more so for low fractions of missing data. For LST, the ranking between missForest and FluxnetEO gap-filling depends on the gap-fraction, missForest consistently produces higher NSE for the lower gap-fractions, FluxnetEO for 40% of samples removed (Fig. 7 left). For Landsat, the NSE of the gap-fill estimates is generally comparable (derived vegetation indices) or better (spectral bands) in FluxnetEO than from missForest (Fig. 7 right). The performance of FluxnetEO is more sensitive to the amounts of missing values than the missForest (Fig. C1 bottom panels). A few more points are of note: For both MODIS and Landsat, the gap-fill estimates of spectral surface reflectance in the visible range (blue, green, red) is less reliable than the one in channels with longer wavelength or derived vegetation indices. The overall gap-fill performance is not satisfactory for Landsat, neither from the FluxnetEO nor from missForest. We did additional tests and found that the signal to noise ratio and the temporal resolution are decisive for the success of the gap-filling. The time series of the average across all subpixels in the Landsat cutout exhibit less noise than the time series of the centre pixel, which also clearly increases the NSE of the artificial gap-fill estimates (Fig. C2, left). FluxnetEO generally performs better on daily than on monthly data (see the lower NSE for MODIS at monthly resolution in Fig. C2, right), which calls for attempts to improve the reliability of FluxnetEO at different temporal resolutions in future releases.

Figure 8 compares the spatial and temporal patterns of Landsat NIR reflectance from FluxnetEO and Moreno-Martínez et al. (2020) across sites and shows a high consistency (panels a,b,d). The largest differences and lowest consistency in both spatial and temporal patterns happen outside the growing season (DJF in large parts of the CONUS, panels b,d,f). This can be expected as NIR reflectance is low during this time of the year, and because the treatment of snow and clouds differs between the products (see time series of one example site in Fig. C8). The temporal correlation of the deviations from the mean seasonality has a bimodal pattern with partly low Pearson correlations of under 0.5 (panel e). The consistency between FluxnetEO and Moreno-Martínez et al. (2020) surface reflectance products generally increases with wavelength, with the lowest agreement for the blue spectral band (Fig. C3, C4, C5, C6, C7).

These benchmarking exercises illustrate important shortcomings but at the same time clearly support the quality of the gap-filling approach proposed by FluxnetEO as being comparable to or slightly higher than independent approaches and products.

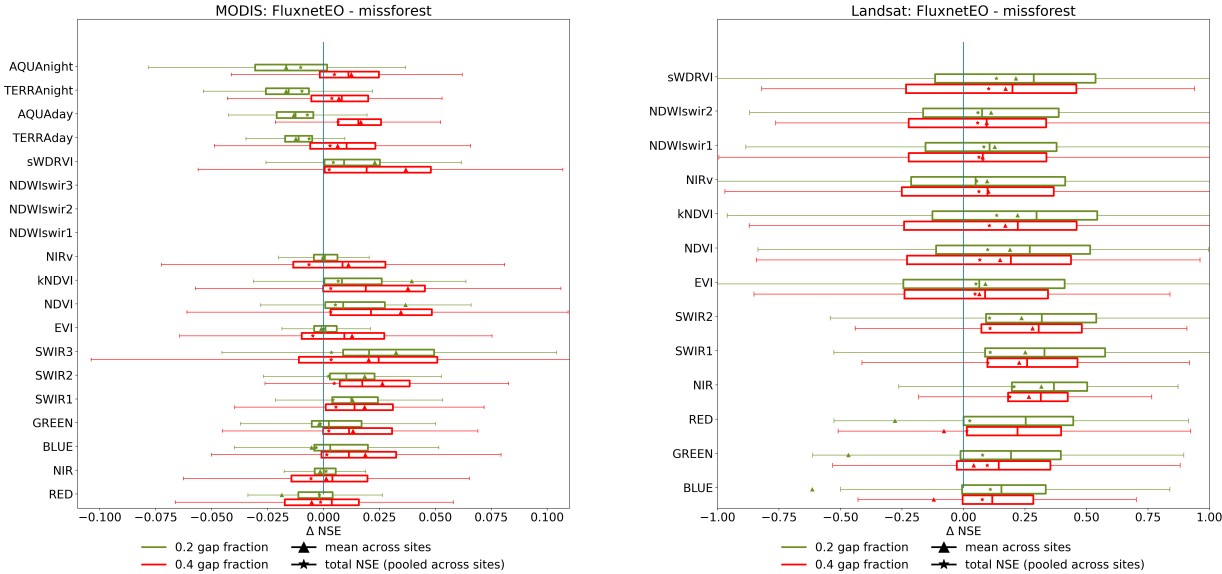

**Figure 7.** Benchmarking in artificial gaps: distribution of NSE per site of the gap-fill estimates in artificial gaps by FluxnetEO compared to missForest within the physical ranges of the indices for 20% and 40% of good quality data removed. For MODIS (left) and Landsat (right), random good quality samples are removed from the tower pixel.

The artificial gaps at random positions in the first experiment might be comparable to those expected from bad inversion or clouds. Removing longer consecutive periods such as during snow periods or persistent cloud cover in rainy seasons is not feasible due to limited consecutive good quality data, so we cannot test the performance for gaps of this type. Compared to missForest, FluxnetEO has the great advantage of being easily scalable to large-scale gridded data products. Compared to the product of Moreno-Martínez et al. (2020) FluxnetEO offers coverage at global sites and is not restricted to the CONUS but lacks the availability of gridded data.

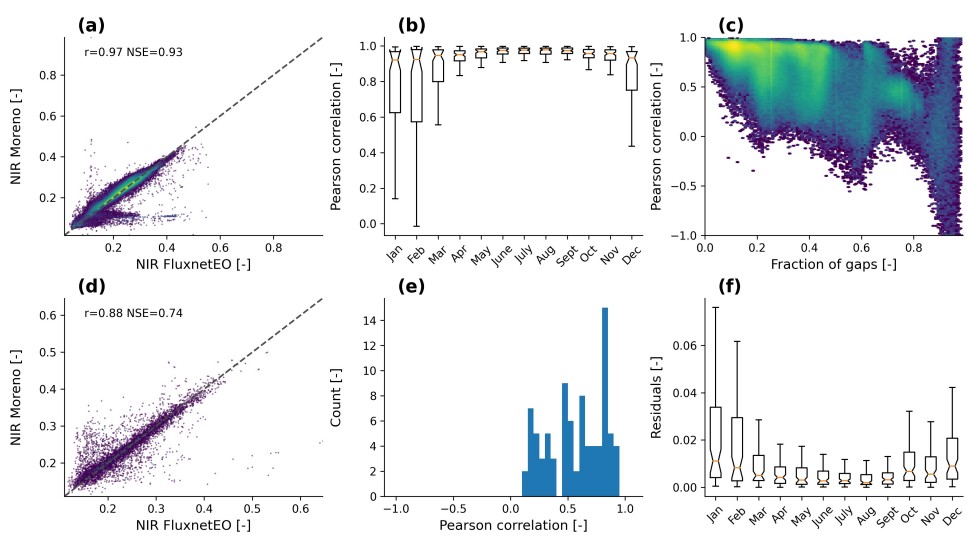

**Figure 8.** Benchmarking Landsat NIR reflectance from FluxnetEO against the product produced by Moreno-Martínez et al. (2020) at EC sites in the CONUS. Each NIR_s,t,p value refers to one site (s), time step (t) and subpixel (p). Comparing spatial patterns: (a) scatterplot of the temporally averaged NIR reflectance (mean_t(NIR_s,p,t), each dot reflects one subpixel and site. (b) Temporal average across years for each month separately and the spatial Pearson correlation across all subpixels in a cutout per site and month cor_p(mean_t-month(NIR FluxnetEO_s,p,t), mean_t-month(NIR Moreno et al_s,p,t)). (c) Temporal correlation in dependence of the amount of missing values in the FluxnetEO product in each subpixel and site (cor_t(NIR FluxnetEO_s,t,p, NIR Moreno_s,t,p). (d-f): Compute a spatial average across all subpixels in a cutout per time step: NIR*_s,t = mean_p(NIR_s,t,p). (d) Temporal Pearson correlation of the spatially averaged NIR (cor_t(NIR FluxnetEO*_s,t, NIR Moreno*_s,t). (e) Pearson correlation of the deviations from the mean seasonal cycle of the spatially averaged time series. (f) Difference between FluxnetEO and Moreno NIR reflectance and their average per month of the year mean_t-month(NIR FluxnetEO*_s,t - NIR Moreno*_s,t). r refers to the Pearson correlation coefficient, NSE to the Nash-Sutcliffe efficiency (Nash and Sutcliffe, 1970).

## 4.4 On the importance of spatial context

In this section, we present different examples of the relevance of spatial context. The type and distribution of the vegetation around a given EC measurement station are not necessarily homogeneous. Instead, clusters of different vegetation or land use types might prevail in different sections of the immediate surroundings of a site. The area that a given flux measurement is representative of (the flux footprint, Schmid, 1997) changes rapidly with wind direction, turbulence conditions, atmospheric stability, and surface resistance (Schmid, 1997; Vesala et al., 2008; Chu et al., 2021). An exact match between the flux footprint and EO data (or a model grid cell) is challenging due to the often unknown or uncertain flux footprints and coarse spatial grid sizes. The scale mismatch is equally important for validation exercises for site-level measurements of surface reflectance (Román et al., 2009; Cescatti et al., 2012), site-level energy-balance closure (Stoy et al., 2013) and model-data integration (Williams et al., 2009). The role that the scale-mismatch between site-level and EO data plays for ecosystem analyses clearly depends on the site and the application. Some applications try to account for the mismatch (Pacheco-Labrador et al., 2017; Wagle et al., 2020); others ignore it and use a custom area around each EC site. Approaches to quantify and account for heterogeneity within a satellite pixel or a certain area around a given site do exist in the literature (Román et al., 2009; Chu et al., 2021; Duveiller et al., 2021) but seem less exploited.

We computed the average flux footprints for every day (MODIS) and month (Landat) around three example EC stations (Majadas de Tietar, ES-LM1, Gebesee, DE-Geb, and Zotino, RU-Zo2). We illustrate how the relationship between EC-derived gross primary productivity (GPP) and EVI as an EO-derived proxy of the same changes according to whether the footprint area is taken into account or custom cutout sizes are chosen. In RU-Zo2, we compare surface temperature inverted from sensible heat flux to LST and illustrate how the pixel sizes relate to the flux footprint area (see details on the data processing in Appendix D).

The site ES-LM1 (El-Madany et al., 2018) is a tree-grass ecosystem. While the trees are evergreen, the herbaceous layer senesces in summer and re-greens in autumn (Luo et al., 2018). The EO cutout includes irrigated agricultural areas north of the flux footprint. These fields are barren in winter and are covered with crops in summer. MODIS and Landsat EVI are strongly negatively correlated to GPP derived from EC in the pixels over agricultural areas, as are the anomalies of EVI and GPP (Fig. D1 a-d). Conversely, high positive correlations prevail across the remaining larger parts of the EO cutouts. Landsat EVI overlaid by the average flux footprint for two example months illustrates that the EC GPP is only representative of the tree-grass ecosystem (Fig. 9e, g). Hence, the spatial representativeness of EO data for EC fluxes might differ strongly depending on which satellite pixels are chosen for the analysis. We computed the average EVI that is representative of the flux footprint (henceforth fpa for footprint area). We compared it with an average EVI weighted with the probability density function of the flux footprint in order to take into account the decreasing influence of subpixels further away from the tower (henceforth fpw for weighted footprint area), as well as with two pragmatic approaches in case a flux footprint is unknown: an EVI average over all subpixels in the cutout with a radius of 2 km (henceforth fex for full extent) or only the single subpixel that contains the tower (cpx for center pixel). The most noticeable difference between the time series for the different intersection methods

is that the full extent (fex) in both Landsat and MODIS EVI is comparatively lower during the winter period (Fig. 9a,c). The agricultural areas contribute to fex, while the footprint intersection methods (fpa and fpw) and the centre pixel (cpx) EVI consistently indicate high greenness in the tree-grass ecosystem.

Gebesee, DE-Geb, is an agricultural site. The common approach in conducting EC measurements is to put the tower in a location where the land use is as homogeneous as possible, to be able attribute fluxes to a targeted ecosystem, e.g. a known crop type. In Gebesee, this was assured for most of the years in the long site history (e.g. Fig. 9h), but not from 2011-2013. In these years, the field was split into two different adjacent crop types that contributed to the measured fluxes (Fig. 9f), raising the risk for pitfalls in the analyses of the fluxes. Also, in situations/ years when the flux footprint represents a single field, additional potential difficulties originate from phenological differences between fields within the EO cutouts (Fig. 9f,h) if not properly matched. For example, the anomalies of both GPP and EVI are only highly correlated with each other in the immediate surroundings of the tower (Fig. D1g-h). Phenological heterogeneity between fields might explain why the EVI averaged over the full cutout (fex) is clearly different from the EVI in the footprint area (fpa, fpw) or the tower pixel (cpx) during the growing season maxima in 2015/16 (Fig. 9b,d). Also, consistently with the GPP, the EVI in the tower pixel indicates slightly later senescence in 2017 than averaged over the footprint area or the full cutout, highlighting considerable effects of a mismatch between the flux footprint and the EO area.

Irrespective of the match between flux footprint and the area that the EVI is representative of, Fig. 9 illustrates the complimentarity between MODIS and Landsat in terms of resolution. Although Landsat offers high spatial detail, the temporal patterns that can be resolved with monthly averages are much coarser than the shorter variations that daily MODIS data can describe. Depending on the application the user of FluxnetEO might choose one or the other.

RU-Zo2, the Zotino tall tower observatory ZOTTO, is located in the taiga-tundra transition zone. The landscape in the proximity of the EC station is a heterogeneous mix of forest, bogs and wetlands. At the tall tower, fluxes are measured at different heights above the canopy. The size of the flux footprint strongly increases with height and the fluxes at the highest level partly represent areas more than 2 km away from the site (Fig. 10b-d). Flux footprints of measurements closer to the canopy are usually much smaller than the MODIS pixel size of 1 km for the LST, but the flux footprints of the higher measurement levels at RU-Zo2 partly integrate over multiple of such pixels. Size and direction of the footprint extents strongly vary over time (note that Fig. 10b-d represent three consecutive days), such that the vegetation types and surface conditions sampled do not only differ between measurement heights but also between days. We compare spaceborne LST AQUA$_{day}$ integrated over the flux footprint area (LST$_{fpa}$) with surface temperature inverted from sensible heat flux measured at the tower for clear-sky days (Fig. 10a, see details about the methods in Appendix D). We observe a tendency of LST$_{fpa}$ at all three measurement heights to be slightly lower than inverted surface temperature under freezing conditions with a notable scatter. For temperatures above 0 °C, the scatter decreases and LST$_{fpa}$ of all three heights is consistently higher than the inverted surface temperature. For the peak surface temperatures during a year (above approximately 285 K), the slope between LST$_{fpa}$ and surface temperature visually decreases, which might indicate significant changes in surface emissivity during the brief peak growing season when

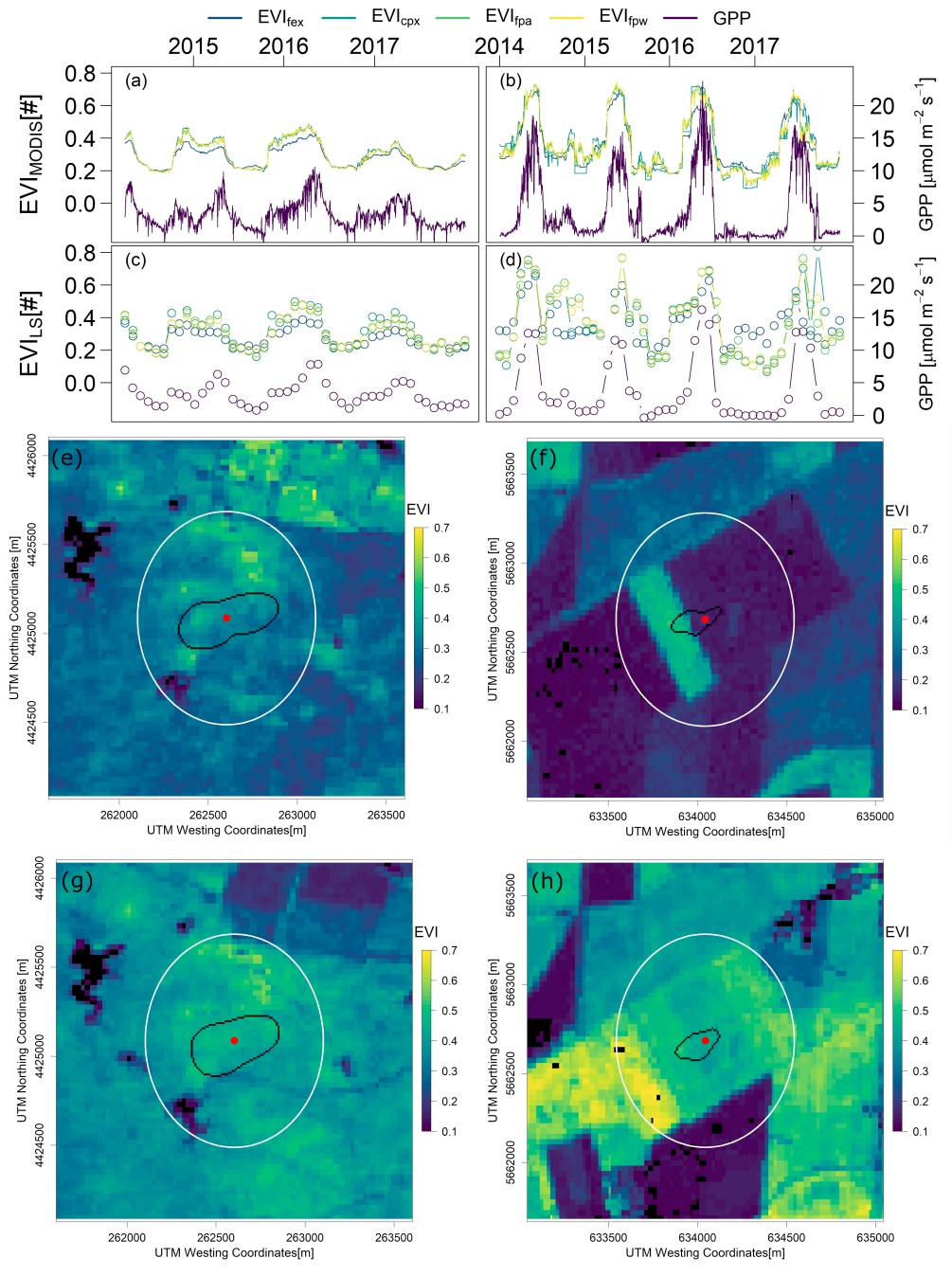

**Figure 9.** Time series of EVI and GPP for ES-LM1 (a,c) and DE-Geb (b,d). MODIS EVI (top row) and Landsat EVI (second row) represent areas with different extents: full extent of the cutout ($EVI_{fex}$), the center pixel that contains a tower ($EVI_{cpx}$), the EVI averaged over the flux footprint area ($EVI_{fpa}$), and the $EVI_{fpa}$ weighted with the flux probability density function ($EVI_{fpw}$). Subplots e-h: Landsat EVI overlaid with the monthly flux footprint (black line) for ES-LM1 in November 2014 (e) and April 2016 (g), and for DE-Geb in February 2012 (f) and February 2016 (h). Non-original low quality EVI values are blacked out. Red circles indicate the location of the EC station, white circle denotes 1 km diameter from the station.

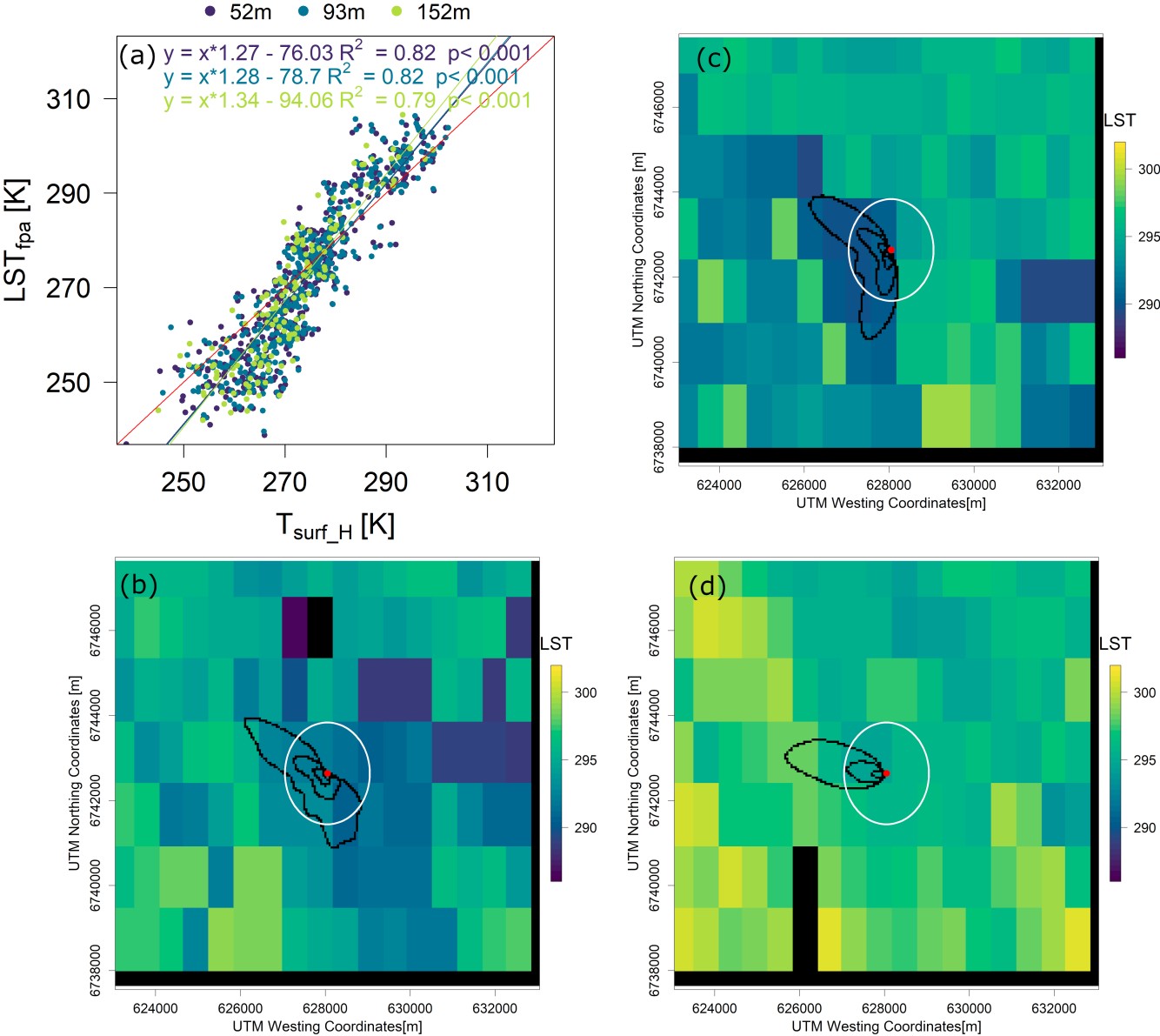

**Figure 10.** Relationship between MODIS AQUA$_{day}$ LST$_{fpa}$ and surface temperature (T$_{surf\_H}$) calculated from the inverted sensible heat flux (details about the methods in Appendix D). The red line represents the 1:1 line. Subplots b to d show example footprints at the three levels (black lines) overlaid on the LST map from May 31st to June 2nd, 2017, respectively. Non-original low quality LST values are blacked out. The white circle indicates the 1 km diameter around the tower.

vegetation extent is highest and the surface has drained from snow melt.

Next to matching the flux footprints with the EO data pixels, spatial context is equally important in studies of vegetation recovery after a disturbance event. The Sky Oaks-Young Stand (US-SO3) is a closed shrubland with less than 2 m tall woody vegetation. The US-SO3 site experienced a fire during the period 2002-2003, followed by regrowth. Landsat allows to observe the impact structure and the spatially very heterogeneous recovery dynamics with remarkable detail (Fig. 11): The fire caused lower than average EVI in large parts of the cutout during the period 2002-2004 (Fig. 11d-f). From 2005 onwards, some patches, particularly the western part of the cutout, appear to have recovered faster from the disturbance than other patches (Fig. 11g). By 2011, EVI has reached pre-fire values in most parts of the area around the site with only small patches as exceptions indicating that regrowth was complete (Fig. 11n). This example illustrates how high spatial resolution EO combined with EC at the site-level can provide complementary insights for better understanding disturbance regimes and the associated recovery dynamics.

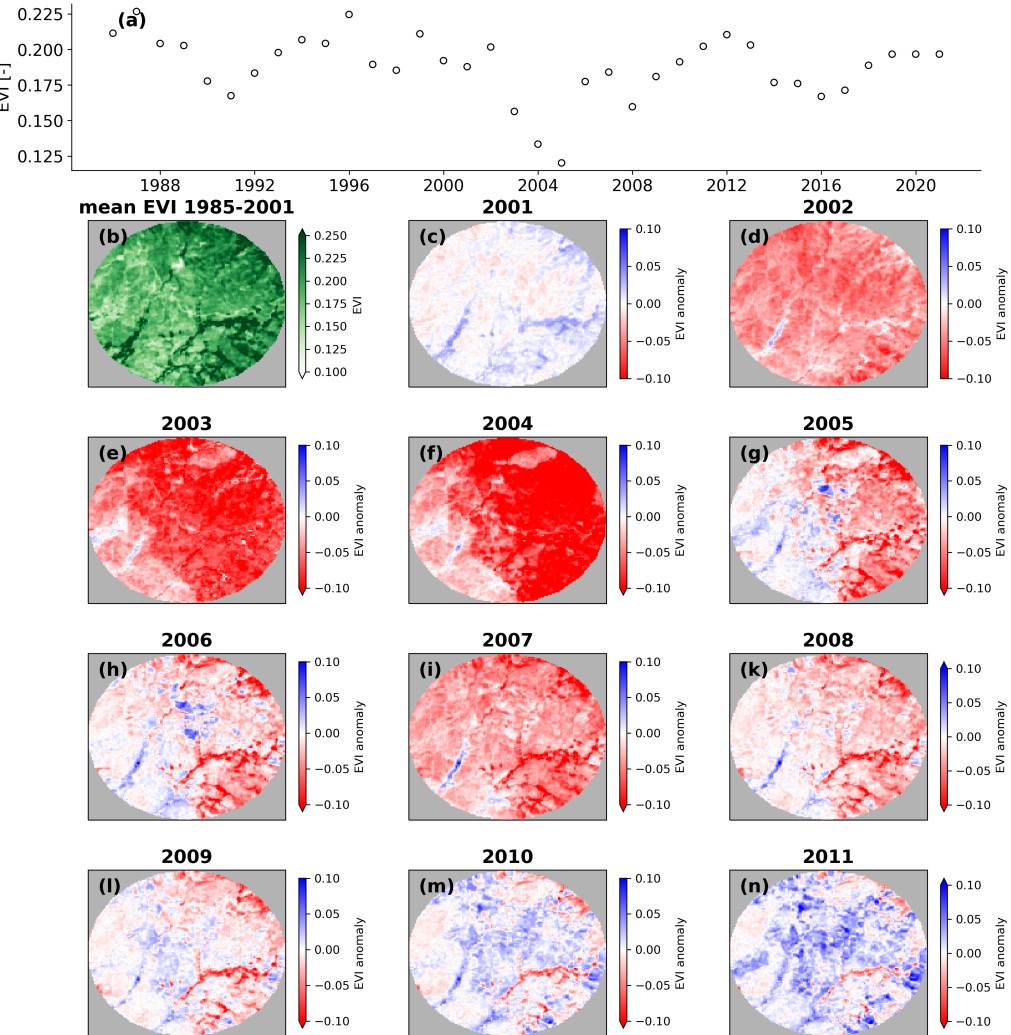

**Figure 11.** Annual EVI dynamics at the site US-SO3 as observed by Landsat. Time series of spatial average annual EVI for the full 4x4km$^2$ cutout (a) and the long-term temporal average spatial patterns of EVI (b). Annual anomalies of EVI for the period 2003-2011 in panels c-n (anomaly $EVI_{year\ n}$ = $EVI_{year\ n}$ - mean($EVI_{1985-2001}$).

## 5    Conclusions

The proposed methods aim at assuring good quality and producing as reliable as possible gap-free estimates of EO-derived
surface reflectance, vegetation indices, and LST for pixels around EC sites, while remaining independent of additional data
sources and being generalisable. Depending on the question/ application at hand, either MODIS or Landsat EO data might
be more suitable with their inherently very diverse spatial and temporal resolutions, reliability of the gap-filling approach and
temporal coverage. The requirements for the strictness of the quality checks and the sophistication of the gap-filling methods
differ by use case. No approach can fit all requirements, but we expect FluxnetEO to offer many opportunities to advance
our understanding of land-atmosphere fluxes for individual sites across regional networks and globally. It helps bridging the
Fluxnet, remote sensing, and modelling communities, and facilitates consistent benchmarking of EO-based flux models of any
kind. We anticipate that this will accelerate our ability to monitor and understand land-atmosphere fluxes across spatial and
temporal scales. For the future, we plan to maintain, update and improve FluxnetEO. This will include extending the time
series to the most recent years, adding EC sites as measurements become available in one of the networks, improving the
processing based on newly identified drawbacks and/ or user needs (e.g., Landsat sensors harmonisation, better performance
also at lower temporal resolutions), and updating to new EO data collections (e.g., Landsat collection 2, integration of Landsat
9). Importantly, forthcoming FluxnetEO versions shall more strongly facilitate complementary usage of multiple missions to
exploit their synergy potential, so that future additions will include further EO products, for example, the Sentinel missions.
Although temporal overlap with most of the EC records is low, it will grow with the lifetime of the different Sentinels because
strong efforts in the EC community target the timely, free, and open distribution of site-level measurements.

*Data availability.*    Data sets are available for open and free usage under ICOS Carbon Portal in separate collections for Landsat (Walther
et al., 2021a, https://meta.icos-cp.eu/objects/koxcGgkI7dDipnTiFyB-RTR5) and for MODIS (Walther et al., 2021b, https://meta.icos-cp.
eu/collections/tEAkpU6UduMMONrFyym5-tUW). Zipped folders package the data by continents and groups of countries. In the zip-
directories, the files are organised by site and in two processing versions: One version contains spatially explicit data fields for each subpixel
in the cutout of 4x4km$^2$ and is denoted by 'subpixel' in the file name. A second version is an average time series per site that represents the
area within 1km radius of the site ('average_cutout'). The inverse distance to the tower serves as weight in the average to account for the fact
that areas farther away from the stations contribute less to the measured fluxes than the immediate surroundings of a site also in the average
of land surface characteristics. In this version, at every time step all valid subpixels closer than 1km to the site are averaged after the quality
checks, and the gap-filling procedure applies to this average time series. The data fields contained in both processing versions are listed in
table 2. Each data field has a complementary data layer ('gapfilltype') with an integer flagging which data point is of original good quality
(=0) or in which gap-filling step a given point has been imputed in the gap-filling procedure (flags >=1). The key to this integer flag is given
in the file attributes. The processing version 'average_cutout' has additional fields that indicate how many valid pixels within 1km of the
tower contributed to the spatial average per time step ('N') and the spatial standard deviation of the vegetation index or LST for the given
time step ('NSTD').

**Table 1. FluxnetEO compared to a selection of other products and services featuring Landsat and MODIS EO data**

| | FluxnetEO version1.0 | ORNL DAAC subsetting tool | Robinson et al. (2017) | Moreno-Martinez et al. (2020) | Sun et al. (2017) |
|---|---|---|---|---|---|
| main service | quality-control & gap-filling | sub-setting | quality-control & gap-filling | quality-control & gap-filling, sensor fusion | quality-control & gap-filling |
| products | MODIS NBAR surface reflectance &, vegetation indices (daily, 500m) MODIS LST (daily, 1km) Landsat surface reflectance & vegetation indices (monthly, 30m) | a range of satellites and sensors, complementary to FluxnetEO | Landsat NDVI (30m, 16-daily) | Landsat surf. reflectance and uncertainties in 6 bands (30m, monthly) | MODIS surface reflectance, albedo and BRDF parameters in land bands and three broad bands (daily, 30arcsec) |
| site selection | 338 eddy-covariance sites (LaThuile, Fluxnet2015, ICOS Drought 2015) | more than 3000 field sites of any kind and network | none, gridded data set for the CONUS | none, gridded data set for the CONUS | none, gridded global data set |
| pre-processing | quality control (retrieval, clouds, snow, water, outlier), gap-filling | none | quality-control (clouds, outliers, snow only partly) gap-filling by user-defined climatology | quality control (clouds, snow, water) gap-filling by climatology and sensor fusion | quality control of BRDF parameters (inversion, cloud, snow, outliers) gap-filling of BRDF parameters |
| cutout size | 4x4km², re-projected to regular grid | 8 x 8 km², native projection | none | none | none |
| site coordinates | verified | coordinates reported from the networks | none | none | |
| length of record | 2000-2020 for MODIS 1987-2017 for Landsat regular updates planned | up to the very recent past (about one month), sensor data are only archived for periods when a site is active | 1984-2019 | 2009-2020 | 2000-2017 |
| file format | netcdf | csv, json | GEOtiff | tiff | hdf |
| access | ICOS Carbon portal | ORNL DAAC (2018) | http://ndvi.ntsg.umt.edu/ | GEE | NASA Earthdata |

**Appendix A: Technical details about the processing of surface reflectance**

In this section we provide all specific technical details necessary to reproduce our processing steps for the surface reflectance of MODIS and Landsat.

The quality control of the MODIS reflectance-based land surface indicators included the following steps:

– Omission of the MCD43A2 BRDF_Albedo_Band_Quality_BandX flags $\geq 3$ for each band to remove bad inversion quality from the surface reflectances.

– The flag Snow_BRDF_Albedo eliminated pixels that contain snow. As the gap-filling procedure used the snow information, a spatially aggregated snow flag was needed for the processing version that averages valid data within 1 km of the tower. For this, we defined the aggregated snow flag as the fraction of subpixels in the cutout that are snow covered. If 530 more than 50% of subpixels have missing snow information for a certain day, the aggregated snow flag is set to missing as well.

– The presence of water in a scene seen by an optical sensor can strongly affect the observation. The BRDF_Albedo_LandWaterType flag allowed to filter for pixels exclusively on land (flag=1). This eliminated all data for many Swiss, Dutch, Italian and Finnish sites which are situated close to water bodies. Inclusion of ocean coastlines and lake shorelines (flag=2) and 535 shallow inland water (flag=3) resulted in reasonable time series at most sites. This came at the cost of having few other sites that were affected by the presence of water. As a trade-off between data availability and quality, we decided to include land-water flags 1-3.

– After the computation of the vegetation indices from the individual spectral bands, an additional check removed possible values of the spectral vegetation indices outside their defined ranges. An outlier filter compared each value to the median 540 of all valid values in temporal windows of 30 days (Papale et al., 2006). A large difference of a given value to the median of its surrounding values indicates a potential outlier. The threshold z as in Papale et al. (2006) was set to 2, and only a less conservative threshold of z=3 acted when more than 20 valid values were available in a given window.

The empirical outlier filter for Landsat slightly differed from the one for MODIS and removed observations in the five highest and lowest percentiles of the median seasonal cycle of an index if they differed more than 75% from their surrounding 545 3-months moving window median. The second criterion was critical in order to preserve observations of disturbance events or recovery dynamics.

Technical details for the gap-filling:

1. The first step is a moving window median to fill short non-snow related gaps. If the entire time series has less than 40% 550 valid data, a given moving window contains both the actual values and the median seasonal cycle for the given time of the year. The median for the moving window refers then to the distribution of both.

2. The second step fills reflectance values with a constant value in the presence of snow (snow flag$\geq$0.1). Partly long periods with missing snow information in the Snow_BRDF_Albedo flag needed special treatment. Some of these gaps appeared systematically in early winter in higher latitudes, so also times of missing snow information are considered as snow covered. However, also during the growing season long periods of missing snow information occur in several sites globally. The following criteria check whether a period that is considered snow covered by high values or missing snow flags is filled with a constant baseline value or not:

   – If a given site has less than 60 days with valid snow coverage (i.e. Snow_BRDF_Albedo=1) in the total record, snow typically does not occur at the site. In this case the gap-filling procedure does not apply this gap-filling step at all for this site.

   – The gap-filling with a constant value only addresses gaps with a minimum length of 20 consecutive days with snow flag missing or 1. This avoids filling very short intermittent snow periods or short gaps in snow information during the growing season.

   – This gap-filling step does not consider gaps due to missing snow information if the median seasonal cycle of snow coverage indicates $\leq$ 5% of snow cover at the given time of the year and the difference between the fill value and the median seasonal cycle is large (i.e. exceeds the 85$^{th}$ percentile of the differences in times of missing snow information).

The constant baseline value that is used to fill snow periods in the time series for a site represents the 3$^{rd}$ percentile of the median seasonal cycle of the spectral vegetation indices. If a given index typically has high values outside the growing season, the baseline value represents the 97$^{th}$ percentile instead. However, if for a given winter the average over the last 5 valid data points at the end of the growing season or over the first 5 valid data points at the beginning of the next growing season is lower than the baseline value (higher than the baseline for indices which are typically high outside the growing season), the baseline takes the value of this average for the given winter (similar to Beck et al., 2007).

3. Linearly scale the median seasonal cycle (MSC) to the time series to fill longer gaps (Verger et al., 2013). Calibration happens in moving temporal windows of 80 days, and application of the scaling in steps of 20 days. In the following x represents a time series of reflectance-based indices, and x* the time series with some of its gaps filled by a scaled MSC.

$$x_{t=k:k+80} = f\,(\,MSC(\,x\,)_{t=k:k+80}\,)$$
$$x^*_{t=k:k+20} = m \cdot MSC(\,x\,)_{t=k:k+20} + n$$

## Appendix B: Technical details about the processing of MODIS LST

In this section we provide all specific technical details necessary to reproduce the processing steps for the MODIS LST.

The empirical filter to remove potential outlier values (Papale et al., 2006) followed the same procedure like for the vegetation indices, but used a constant z-value of 1.5 as it provided the best trade-off between filter success, false positives and false

negatives.

Estimates of LST in data gaps originate from the following steps:

– In contrast to the procedure for the reflectance-based vegetation indices, the distribution of values in the temporal windows of 8 days is not supplied by the median seasonal cycle in case of low data availability. The moving window median was not applied for windows with less than three valid values.

– Filling by linearly scaling the median seasonal shift between any two of the four MODIS LST time series to each other (Crosson et al., 2012; Li et al., 2018). The following explains this gap-filling step for $TERRA_{day}$ as the 'imputed' time

series:

    1. Compute the shift between $TERRA_{day}$ and $AQUA_{day}$ ($\Delta(TERRA_{day}, AQUA_{day})$) and obtain the MSC of the shift : MSC( $\Delta(TERRA_{day}, AQUA_{day})$ ).

    2. Linearly scale the MSC of the shift to the shift itself in temporal windows of 80 days (provided a minimum of 10 valid values in a given window). Apply the scaling in windows and steps of 20 days to obtain estimates of the

shift ($\Delta(TERRA_{day}, AQUA_{day})$)*) from its MSC where it is missing.

$\Delta(TERRA_{day}, AQUA_{day})_{t=k:k+80} = f ( MSC( \Delta(TERRA_{day}, AQUA_{day}) )_{t=k:k+80} )$

$\Delta(TERRA_{day}, AQUA_{day}))^*_{t=k:k+20} = m \cdot MSC( \Delta(TERRA_{day}, AQUA_{day}) )_{t=k:k+20} + n.$

    3. Add the scaled average shift to the $AQUA_{day}$ to obtain an estimate of $TERRA_{day}*[AQUA_{day}]$ that can fill gaps in $TERRA_{day}$.

$TERRA_{day}*_{t=k:k+20}[AQUA_{day}] = AQUA_{day t=k:k+20} + \Delta(TERRA_{day}, AQUA_{day}))^*_{t=k:k+20}$

Analogously to $TERRA_{day}*[AQUA_{day}]$, also the night-time LST observations contributed to estimate $TERRA_{day}*[TERRA_{night}]$ and $TERRA_{day}*[AQUA_{night}]$. All three estimates $TERRA_{day}*[AQUA_{day}]$, $TERRA_{day}*[TERRA_{night}]$ and $TERRA_{day}*[AQUA_{night}]$, served to fill gaps in $TERRA_{day}$, namely in the order of increasing standard deviation of the differences between valid $TERRA_{day}$ and each of the three estimated $TERRA_{day}*$.

The procedure analogously filled $AQUA_{day}$, $TERRA_{night}$ and $AQUA_{night}$ accordingly using valid observations of the remaining three, respectively.

– Linearly scale the valid LST observations of each of the four data streams to their own median annual cycle in temporal windows. As in step 2, the calibration happened in temporal windows of 80 days, while the scaling was applied in windows of 20 days. Exemplarily for $TERRA_{day}$:

$TERRA_{day t=k:k+80} = f ( MSC( TERRA_{day} )_{t=k:k+80} )$

$TERRA_{day}*_{t=k:k+20} = m \cdot MSC( TERRA_{day} )_{t=k:k+20} + n$

## Appendix C: Details about the benchmarking exercises

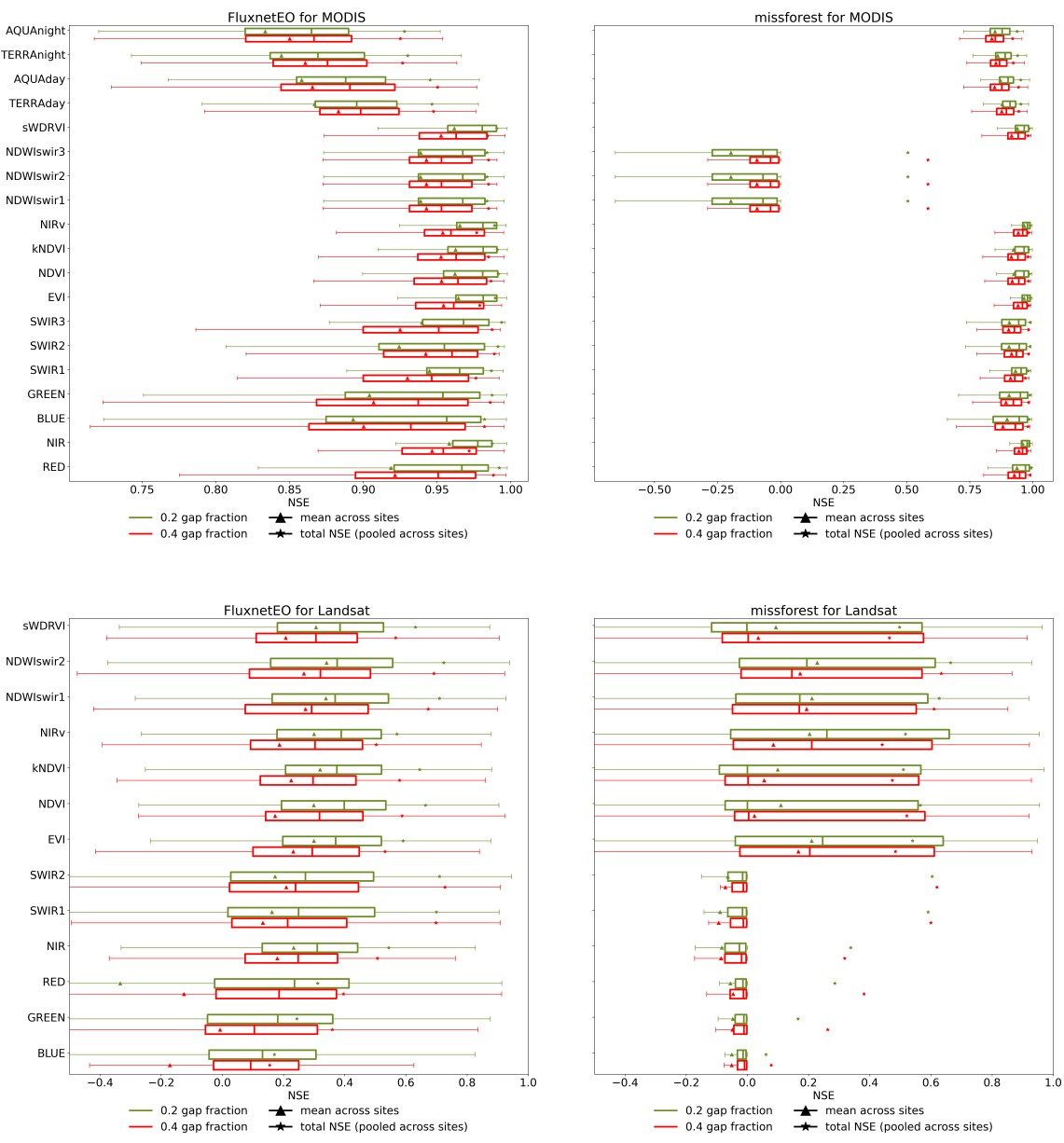

**Figure C1.** Benchmarking in artificial gaps: distribution of NSE per site of the gap-fill estimates in artificial gaps by FluxnetEO (left) and missForest (right) within the physical ranges of the indices for 20% and 40% of good quality data removed. For MODIS (top) and Landsat (bottom), random good quality samples are removed from the tower pixel. Note the different x-axis limits.

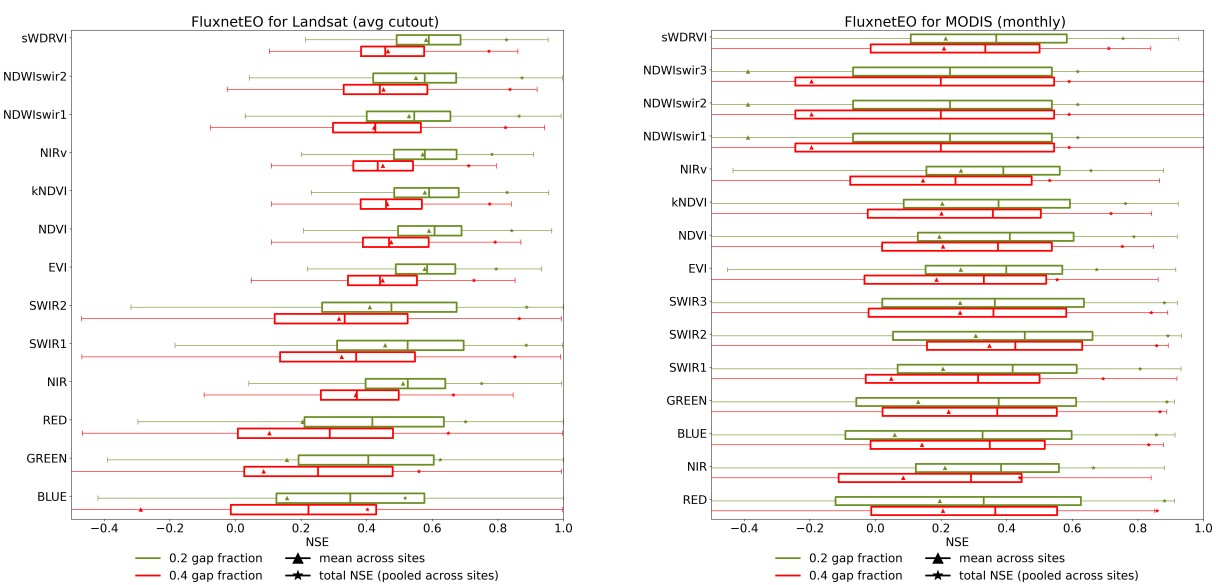

**Figure C2.** Benchmarking in artificial gaps: distribution of NSE per site of the gap-fill estimates in artificial gaps by FluxnetEO. 20% and 40% of data were removed and gapfilled. Left: Landsat time series of the average reflectance/ vegetation index across the whole cutout. Right: the centre pixel of MODIS data aggregated to monthly temporal resolution.

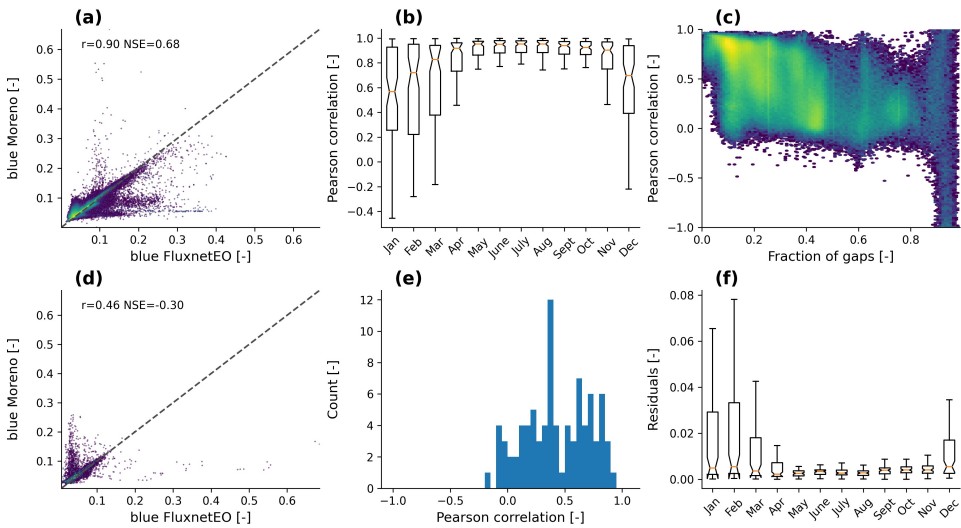

**Figure C3.** Benchmarking Landsat reflectance in the blue spectral band from FluxnetEO against the product produced by Moreno-Martínez et al. (2020) at EC sites in the CONUS. Each reflectance_s,t,p value refers to one site (s), time step (t) and subpixel (p). Comparing spatial patterns: (a) scatterplot of the temporally averaged reflectance (mean(reflectance_s,p)_t, each dot reflects one subpixel and site. (b) Spatial Pearson correlation across all subpixels in a cutout per site of the average grouped by month. (c) Temporal correlation in dependence of the amount of missing values in each subpixel and site. (d-f): Compute a spatial average across all subpixels in a cutout per time step. (d) Temporal Pearson correlation of the spatial average. (e) Pearson correlation of the deviations from the mean seasonal cycle of the spatially averaged time series. (f) Difference between FluxnetEO and Moreno reflectance and their average per month of the year. r refers to the Pearson correlation coefficient, mef to the Nash-Sutcliffe efficiency (Nash and Sutcliffe, 1970).

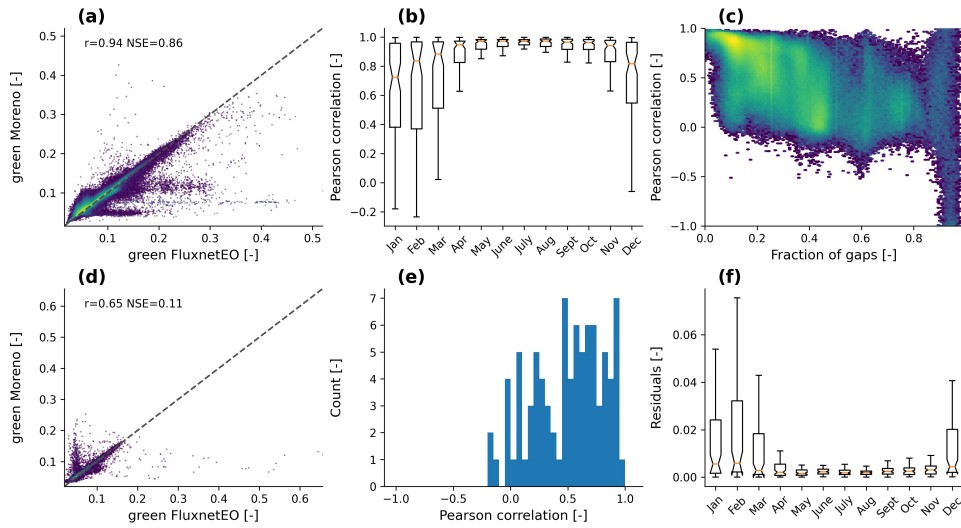

**Figure C4.** Same like Fig. C3 for the green spectral band.

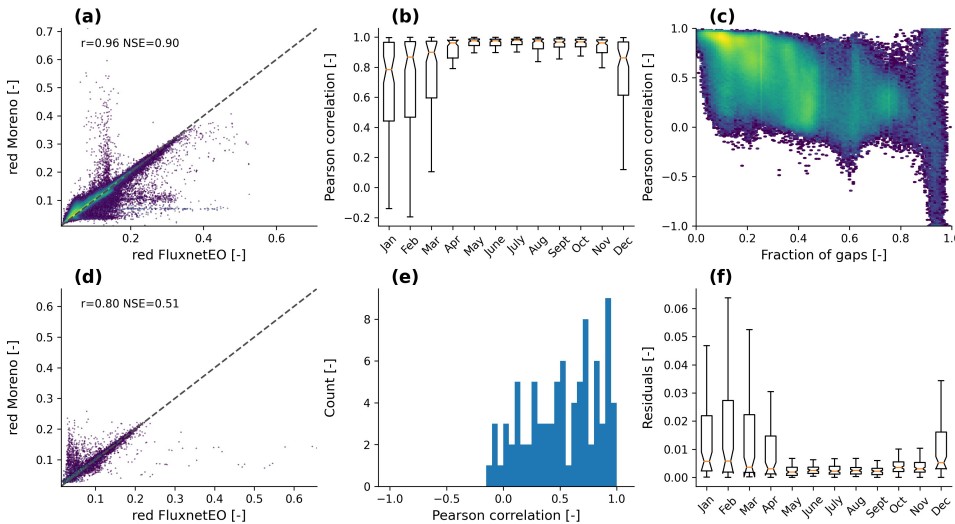

**Figure C5.** Same like Fig. C3 for the red spectral band.

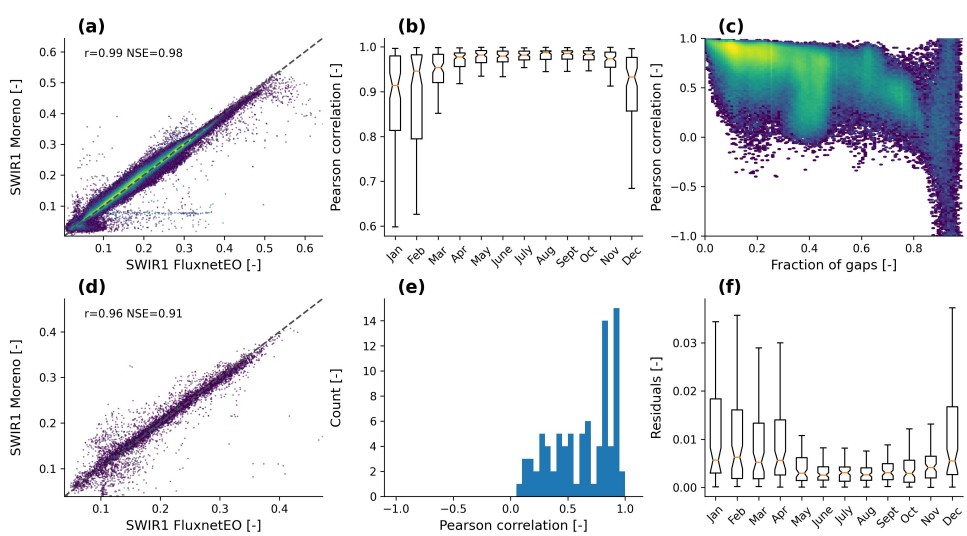

**Figure C6.** Same like Fig. C3 for the first shortwave infrared spectral band.

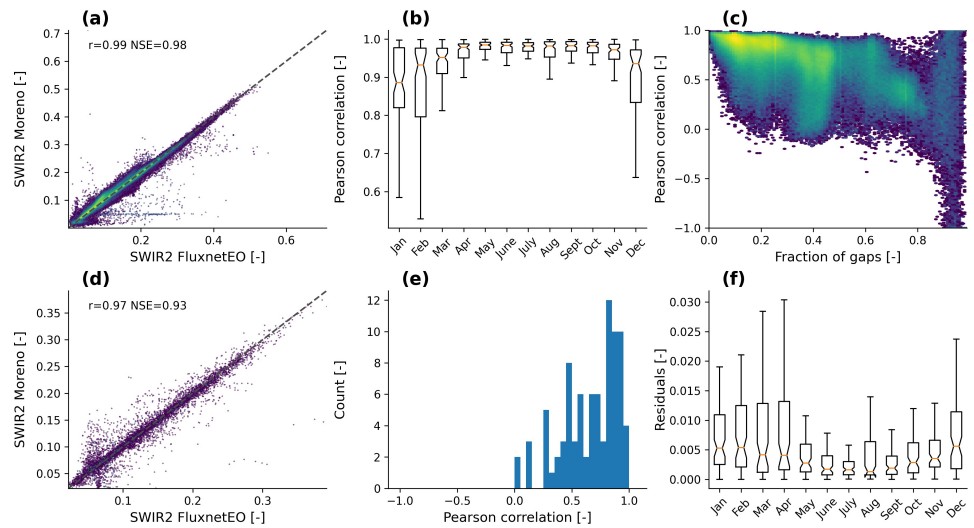

**Figure C7.** Same like Fig. C3 for the second shortwave infrared spectral band.

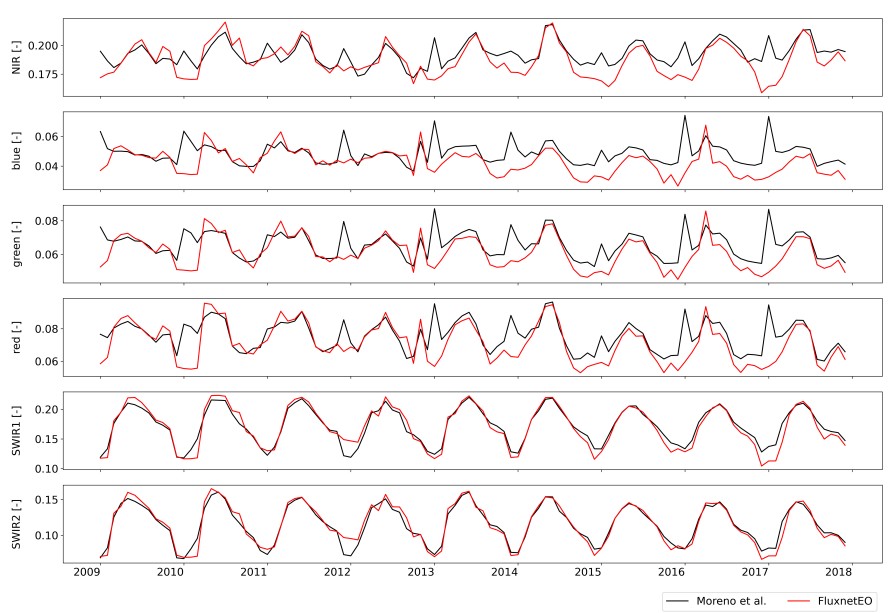

**Figure C8.** Example site US-Fmf: Comparing the gapfilled surface reflectance products in spectral channels.

## Appendix D: Details about the analysis of spatial context

For the analysis at DE-Geb and ES-LM1 we used night-time partitioned GPP (Reichstein et al., 2005) with the mean of the variable u⋆-threshold (GPP_NT_VUT_MEAN) from the Drought 2018 Team and ICOS Ecosystem Thematic Centre (2020) data release (Migliavacca et al., 2020; ICOS Ecosystem Thematic Centre and Gebesee, 2019). We computed the actual flux footprints after Kljun et al. (2015) from ICOS drought 2018 data (Drought 2018 Team and ICOS Ecosystem Thematic Centre, 2020) using the R-code version (V1.41) of the FFP-tool. As a flux footprint for the intersection with EVI we define the area that contributes 80% to the flux footprint probability density function (80% isoline of the monthly/daily cumulative flux footprint for Landsat and MODIS, respectively).

Flux footprint calculation followed the same procedure for the three measurement heights at RU-Zo2. Surface temperature was inverted from sensible heat flux and meteorological variables (Knauer et al., 2018) following equation:

$$\mathrm{Tsurf = Tair + H} / (\rho * c_p * Ga_h)$$

with Tair the air temperature at measurement height (K), H the sensible heat flux (W m-2), $\rho$ the density of air (kg m-3), $c_p$ the specific heat capacity of the air (J kg-1 K-1), and $G_{ah}$ the aerodynamic conductance to heat (m s-1). $G_{ah}$ is defined as $G_{ah} = 1 / (Ra_m + Rb_h)$, with the aerodynamic resistance to momentum $Ra_m = u/ustar^2$ and the canopy boundary layer resistance for heat $Rb_h = 6.2*ustar^{-2/3}$. As the inverted surface temperature was compared to LST $AQUA_{day}$, the average of half-hourly sensible heat flux of the nominal overpass time at 1.30pm $\pm$ 1.5 hours was taken. Only days with good quality in both the LST and and sensible heat flux are used according to the following criteria: i) more than 90% of the EO cutout have valid (i.e. non-gapfilled) values which restricts the comparison to clear-sky conditions, and ii) at least 50% of the half-hourly long-wave fluxes and all meteorological data in a given day are of good quality. A larger cutout of 5x5 km$^2$ was extracted for MODIS LST to fully cover also the extent of the flux footprint of the highest measurement level, but is used only for illustrative purposes and not in the data provided in the FluxnetEO collections.

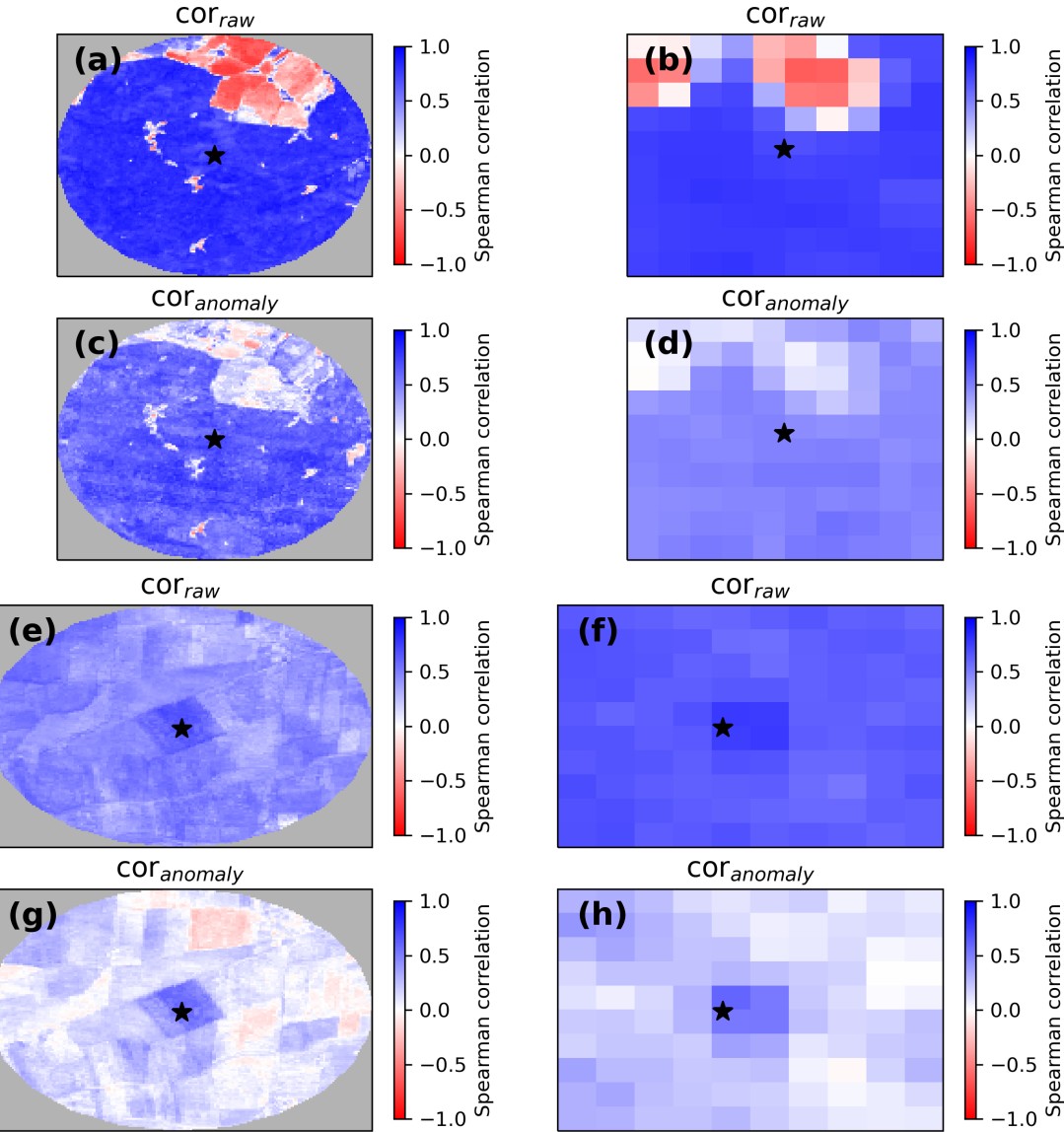

**Figure D1.** Spearman correlation between EVI and GPP using monthly Landsat (a, c, e, g) and daily MODIS (b, d, f, h) data for ES-LM1 (a-d) and DE-Geb (e-h) Fluxnet sites. The correlation estimates were computed on the raw time series (a, b, e, f) and on the anomalies (c, d, g, h).

**Appendix E:  Site selection**

**Table E1.** Sites in FluxnetEO product version 1.0: site codes and coordinates (latitude in degree N, longitude in degree E, rounded to 4 decimals). Site codes including a * indicate sites for which currently only MODIS data are provided.

| site code | latitude, longitude | site code | latitude, longitude |
|---|---|---|---|
| AR-SLu | -33.4648, -66.4598 | AR-Vir | -28.2395, -56.1886 |
| AT-Neu | 47.1167, 11.3175 | AU-ASM | -22.283, 133.249 |
| AU-Ade | -13.0769, 131.1178 | AU-Cpr | -34.0021, 140.5891 |
| AU-Cum | -33.6152, 150.7236 | AU-DaP | -14.0633, 131.3181 |
| AU-DaS | -14.1593, 131.3881 | AU-Dry | -15.2588, 132.3706 |
| AU-Emr | -23.8587, 148.4746 | AU-Fog | -12.5452, 131.3072 |
| AU-Gin | -31.3764, 115.7138 | AU-How | -12.4943, 131.1523 |
| AU-RDF | -14.5636, 132.4776 | AU-Rob | -17.1175, 145.6301 |
| AU-TTE | -22.287, 133.64 | AU-Tum | -35.6566, 148.1517 |
| AU-Wac | -37.4259, 145.1878 | AU-Whr | -36.6732, 145.0294 |
| AU-Wom | -37.4222, 144.0944 | AU-Ync | -34.9883, 146.2916 |
| BE-Bra | 51.3076, 4.5198 | BE-Lon | 50.5516, 4.7462 |
| BE-Vie | 50.3049, 5.9981 | BR-Ban | -9.8244, -50.1591 |
| BR-Cax | -1.7197, -51.459 | BR-Ji2 | -10.0832, -61.9309 |
| BR-Sa1 | -2.8567, -54.9589 | BR-Sa2 | -3.0119, -54.5365 |
| BR-Sa3 | -3.018, -54.9714 | BR-Sp1 | -21.6195, -47.6499 |
| BW-Ma1 | -19.9165, 23.5603 | CA-Ca1 | 49.8673, -125.3336 |
| CA-Ca2 | 49.8705, -125.2909 | CA-Ca3 | 49.5346, -124.9004 |
| CA-Gro | 48.2167, -82.1556 | CA-Let | 49.7093, -112.9402 |
| CA-Man | 55.8796, -98.4808 | CA-Mer | 45.4094, -75.5186 |
| CA-NS1 | 55.8792, -98.4839 | CA-NS2 | 55.9058, -98.5247 |
| CA-NS3 | 55.9117, -98.3822 | CA-NS4 | 55.9144, -98.3806 |
| CA-NS5 | 55.8631, -98.485 | CA-NS6 | 55.9167, -98.9644 |
| CA-NS7 | 56.6358, -99.9483 | CA-Oas | 53.6289, -106.1978 |
| CA-Obs | 53.9872, -105.1178 | CA-Ojp | 53.9163, -104.692 |
| CA-Qcu | 49.2671, -74.0365 | CA-Qfo | 49.6925, -74.3421 |
| CA-SF1 | 54.485, -105.8176 | CA-SF2 | 54.2539, -105.8775 |
| CA-SF3 | 54.0916, -106.0053 | CA-SJ1 | 53.908, -104.656 |
| CA-SJ2 | 53.945, -104.649 | CA-SJ3 | 53.8758, -104.6453 |
| CA-TP1 | 42.6609, -80.5595 | CA-TP2 | 42.7744, -80.4588 |
| CA-TP3 | 42.7068, -80.3483 | CA-TP4 | 42.7102, -80.3574 |
| CA-TPD | 42.6353, -80.5577 | CA-WP1 | 54.9538, -112.467 |
| CA-WP3 | 54.47, -113.32 | CG-Tch | -4.2892, 11.6564 |
| CH-Aws | 46.5832, 9.7904 | CH-Cha | 47.2102, 8.4104 |
| CH-Dav | 46.8153, 9.8559 | CH-Fru | 47.1158, 8.5378 |

| site code | latitude, longitude | site code | latitude, longitude |
|-----------|---------------------|-----------|---------------------|
| CH-Lae | 47.4781, 8.365 | CH-Oe1 | 47.2858, 7.7319 |
| CH-Oe2 | 47.2863, 7.7343 | CN-Anh | 33.0, 117.0 |
| CN-Bed | 39.5306, 116.252 | CN-Cha | 42.4025, 128.0958 |
| CN-Cng | 44.5934, 123.5092 | CN-Dan | 30.4978, 91.0664 |
| CN-Din | 23.1733, 112.5361 | CN-Do1 | 31.5167, 121.961 |
| CN-Do2 | 31.5847, 121.903 | CN-Do3 | 31.5169, 121.972 |
| CN-Du1 | 42.0456, 116.671 | CN-Du2 | 42.0467, 116.2836 |
| CN-Du3 | 42.0551, 116.2809 | CN-HaM | 37.37, 101.18 |
| CN-Hny | 29.31, 112.51 | CN-Ku1 | 40.5383, 108.694 |
| CN-Ku2 | 40.3808, 108.549 | CN-Qia | 26.734, 115.0663 |
| CN-Sw2 | 41.7902, 111.8971 | CN-Xi1 | 43.5458, 116.6778 |
| CZ-BK1 | 49.5021, 18.5369 | CZ-BK2* | 49.4944, 18.5428 |
| CZ-Lnz | 48.6816, 16.9464 | CZ-RAJ | 49.4437, 16.6965 |
| CZ-Stn | 49.036, 17.9699 | CZ-wet | 49.0246, 14.7704 |
| DE-Akm | 53.8662, 13.6834 | DE-Bay | 50.1419, 11.8669 |
| DE-Geb | 51.0997, 10.9146 | DE-Gri | 50.95, 13.5126 |
| DE-Hai | 51.0792, 10.453 | DE-Har | 47.9344, 7.601 |
| DE-HoH | 52.0853, 11.2192 | DE-Hte | 54.2103, 12.1761 |
| DE-Hzd | 50.9638, 13.4898 | DE-Kli | 50.8931, 13.5224 |
| DE-Lkb | 49.0996, 13.3047 | DE-Lnf | 51.3282, 10.3678 |
| DE-Meh | 51.2753, 10.6555 | DE-Obe | 50.7867, 13.7213 |
| DE-RuR | 50.6219, 6.3041 | DE-RuS | 50.8659, 6.4471 |
| DE-RuW | 50.5049, 6.331 | DE-Seh | 50.8706, 6.4497 |
| DE-SfN | 47.8064, 11.3275 | DE-Spw | 51.8922, 14.0337 |
| DE-Tha | 50.9626, 13.5652 | DE-Wet | 50.4535, 11.4575 |
| DE-Zrk | 53.8759, 12.889 | DK-Eng | 55.6905, 12.1918 |
| DK-Fou | 56.4842, 9.5872 | DK-Lva | 55.6833, 12.0833 |
| DK-Ris | 55.5303, 12.0972 | DK-Sor | 55.4859, 11.6446 |
| ES-Abr | 38.7018, -6.7859 | ES-Amo | 36.8336, -2.2523 |
| ES-ES1 | 39.346, -0.3188 | ES-ES2 | 39.2756, -0.3153 |
| ES-LJu | 36.9266, -2.7521 | ES-LM1 | 39.9427, -5.7787 |
| ES-LM2 | 39.9346, -5.7759 | ES-LMa | 39.9415, -5.7734 |
| ES-LgS | 37.0979, -2.9658 | ES-Ln2 | 36.9695, -3.4758 |
| ES-VDA | 42.1522, 1.4485 | FI-Hyy | 61.8474, 24.2948 |
| FI-Jok | 60.8986, 23.5134 | FI-Kaa | 69.1406, 27.2698 |
| FI-Let | 60.6418, 23.9595 | FI-Lom | 67.9972, 24.2092 |

| site code | latitude, longitude | site code | latitude, longitude |
|-----------|---------------------|-----------|---------------------|
| FI-Sii | 61.8326, 24.1928 | FI-Sod | 67.3624, 26.6386 |
| FI-Var | 67.7549, 29.61 | FR-Aur | 43.5497, 1.1061 |
| FR-Bil | 44.4937, -0.9561 | FR-EM2 | 49.8721, 3.0206 |
| FR-Fon | 48.4764, 2.7801 | FR-Gri | 48.8442, 1.9519 |
| FR-Hes | 48.6741, 7.0646 | FR-LBr | 44.7171, -0.7693 |
| FR-Lam | 43.4965, 1.2378 | FR-Lq1 | 45.6431, 2.7358 |
| FR-Lq2 | 45.6392, 2.737 | FR-Pue | 43.7413, 3.5957 |
| GF-Guy | 5.2788, -52.9249 | GH-Ank | 5.2685, -2.6942 |
| GL-NuF* | 64.1308, -51.3861 | GL-ZaF | 74.4814, -20.5545 |
| GL-ZaH | 74.4733, -20.5503 | HU-Bug | 46.6911, 19.6013 |
| HU-Mat | 47.8469, 19.726 | ID-Pag | 2.345, 114.036 |
| IE-Ca1 | 52.8588, -6.9181 | IE-Dri | 51.9867, -8.7518 |
| IL-Yat* | 31.345, 35.052 | IS-Gun | 63.8333, -20.2167 |
| IT-Amp | 41.9041, 13.6052 | IT-BCi | 40.5238, 14.9574 |
| IT-Bon | 39.4778, 16.5347 | IT-CA1 | 42.3804, 12.0266 |
| IT-CA2 | 42.3772, 12.026 | IT-CA3 | 42.38, 12.0222 |
| IT-Col | 41.8494, 13.5881 | IT-Cp2 | 41.7043, 12.3573 |
| IT-Cpz | 41.7052, 12.3761 | IT-Isp | 45.8126, 8.6336 |
| IT-LMa | 45.1526, 7.5826 | IT-La2 | 45.9542, 11.2853 |
| IT-Lav | 45.9562, 11.2813 | IT-Lec | 43.3036, 11.2698 |
| IT-Lsn | 45.7405, 12.7503 | IT-MBo | 46.0147, 11.0458 |
| IT-Mal | 46.114, 11.7033 | IT-Noe | 40.6062, 8.1512 |
| IT-Non | 44.6902, 11.0911 | IT-PT1 | 45.2009, 9.061 |
| IT-Pia | 42.5839, 10.0784 | IT-Ren | 46.5869, 11.4337 |
| IT-Ro1 | 42.4081, 11.93 | IT-Ro2 | 42.3903, 11.9209 |
| IT-SR2 | 43.732, 10.291 | IT-SRo | 43.7279, 10.2844 |
| IT-Tor | 45.8444, 7.5781 | JP-MBF | 44.3842, 142.3186 |
| JP-Mas | 36.054, 140.0269 | JP-SMF | 35.2617, 137.0786 |
| JP-Tak | 36.1462, 137.423 | JP-Tom | 42.7395, 141.5149 |
| MY-PSO | 2.973, 102.3062 | NL-Ca1 | 51.971, 4.927 |
| NL-Haa | 52.0036, 4.8056 | NL-Hor | 52.2404, 5.0713 |
| NL-Lan | 51.9536, 4.9029 | NL-Loo | 52.1666, 5.7436 |
| NL-Lut | 53.3989, 6.356 | PA-SPn | 9.3181, -79.6346 |
| PA-SPs | 9.3138, -79.6314 | PL-Wet | 52.7622, 16.3094 |
| PT-Esp | 38.6394, -8.6018 | PT-Mi1 | 38.5406, -8.0001 |
| PT-Mi2 | 38.4765, -8.0246 | RU-Che | 68.613, 161.3414 |
| RU-Cok | 70.8291, 147.4943 | RU-Fy2 | 56.4476, 32.9019 |

| site code | latitude, longitude | site code | latitude, longitude |
|---|---|---|---|
| RU-Fyo | 56.4615, 32.9221 | RU-Ha1 | 54.7252, 90.0022 |
| RU-Ha3 | 54.7046, 89.0778 | RU-Sam | 72.3738, 126.4958 |
| RU-SkP | 62.255, 129.168 | RU-Tks | 71.5943, 128.8878 |
| RU-Vrk | 67.0547, 62.9405 | RU-Zot | 60.8008, 89.3508 |
| SD-Dem | 13.2829, 30.4783 | SE-Abi | 68.3624, 18.7948 |
| SE-Deg | 64.182, 19.5565 | SE-Htm | 56.0976, 13.419 |
| SE-Lnn* | 58.3406, 13.1018 | SE-Nor | 60.0865, 17.4795 |
| SE-Ros* | 64.1725, 19.738 | SE-Sk2 | 60.1297, 17.8401 |
| SE-St1 | 68.3541, 19.0503 | SE-Svb* | 64.2561, 19.7745 |
| SJ-Adv | 78.186, 15.923 | SJ-Blv | 78.9216, 11.8311 |
| SK-Tat | 49.1208, 20.1635 | SN-Dhr | 15.4028, -15.4322 |
| UK-ESa | 55.9069, -2.8586 | UK-Gri | 56.6072, -3.7981 |
| UK-Ham | 51.1535, -0.8583 | UK-PL3 | 51.45, -1.2667 |
| UK-Tad | 51.2071, -2.8286 | US-AR1 | 36.4267, -99.42 |
| US-AR2 | 36.6358, -99.5975 | US-ARM | 36.6058, -97.4888 |
| US-ARb | 35.5497, -98.0402 | US-ARc | 35.5465, -98.04 |
| US-Atq | 70.4696, -157.4089 | US-Aud | 31.5907, -110.5104 |
| US-Bar | 44.0646, -71.2881 | US-Bkg | 44.3453, -96.8362 |
| US-Blo | 38.8953, -120.6328 | US-Bn2 | 63.9198, -145.3782 |
| US-Bn3 | 63.9227, -145.7442 | US-Bo1 | 40.0062, -88.2904 |
| US-Bo2 | 40.009, -88.29 | US-Brw | 71.3225, -156.6092 |
| US-CRT | 41.6285, -83.3471 | US-CaV | 39.0633, -79.4208 |
| US-Cop | 38.09, -109.39 | US-Dk3 | 35.9782, -79.0942 |
| US-FPe | 48.3077, -105.1019 | US-FR2 | 29.9495, -97.9962 |
| US-Fmf | 35.1426, -111.7273 | US-Fuf | 35.089, -111.762 |
| US-Fwf | 35.4454, -111.7718 | US-GBT | 41.3658, -106.2397 |
| US-GLE | 41.3665, -106.2399 | US-Goo | 34.2547, -89.8735 |
| US-Ha1 | 42.5378, -72.1715 | US-Ho1 | 45.2041, -68.7402 |
| US-Ho2 | 45.2091, -68.747 | US-IB1 | 41.8593, -88.2227 |
| US-IB2 | 41.8406, -88.241 | US-Ivo | 68.4865, -155.7503 |
| US-KS1 | 28.4583, -80.6709 | US-KS2 | 28.6086, -80.6715 |
| US-LWW | 34.9604, -97.9789 | US-Lin | 36.3566, -119.8423 |
| US-Los | 46.0827, -89.9792 | US-MMS | 39.3232, -86.4131 |
| US-MOz | 38.7441, -92.2 | US-Me1 | 44.5794, -121.5 |
| US-Me2 | 44.4523, -121.5574 | US-Me3 | 44.3154, -121.6078 |
| US-Me4 | 44.4992, -121.6224 | US-Me5 | 44.4372, -121.5668 |

| site code | latitude, longitude | site code | latitude, longitude |
|---|---|---|---|
| US-Me6 | 44.3233, -121.6078 | US-Myb | 38.0498, -121.7651 |
| US-NC1 | 35.8118, -76.7119 | US-NR1 | 40.0329, -105.5464 |
| US-Ne1 | 41.1651, -96.4766 | US-Ne2 | 41.1649, -96.4701 |
| US-Ne3 | 41.1797, -96.4397 | US-ORv | 40.0201, -83.0183 |
| US-Oho | 41.5545, -83.8438 | US-PFa | 45.9459, -90.2723 |
| US-Prr | 65.1237, -147.4876 | US-SO2 | 33.3738, -116.6228 |
| US-SO3 | 33.3771, -116.6226 | US-SO4 | 33.3845, -116.6406 |
| US-SP1 | 29.7381, -82.2188 | US-SP2 | 29.7648, -82.2448 |
| US-SP3 | 29.7548, -82.1633 | US-SRC | 31.9083, -110.8395 |
| US-SRG | 31.7894, -110.8277 | US-SRM | 31.8214, -110.8661 |
| US-Sta | 41.3966, -106.8024 | US-Syv | 46.242, -89.3477 |
| US-Ton | 38.4316, -120.966 | US-Tw1 | 38.1074, -121.6469 |
| US-Tw2 | 38.1047, -121.6433 | US-Tw3 | 38.1159, -121.6467 |
| US-Tw4 | 38.103, -121.6414 | US-Twt | 38.1087, -121.653 |
| US-UMB | 45.5598, -84.7138 | US-UMd | 45.5625, -84.6975 |
| US-Var | 38.4133, -120.9507 | US-WBW | 35.9588, -84.2874 |
| US-WCr | 45.8059, -90.0799 | US-WPT | 41.4646, -82.9962 |
| US-Whs | 31.7438, -110.0522 | US-Wi0 | 46.6188, -91.0814 |
| US-Wi1 | 46.7305, -91.2329 | US-Wi2 | 46.6869, -91.1528 |
| US-Wi3 | 46.6347, -91.0987 | US-Wi4 | 46.7393, -91.1663 |
| US-Wi5 | 46.6531, -91.0858 | US-Wi6 | 46.6249, -91.2982 |
| US-Wi7 | 46.6491, -91.0693 | US-Wi8 | 46.7223, -91.2524 |
| US-Wi9 | 46.6188, -91.0814 | US-Wkg | 31.7365, -109.9419 |
| US-Wrc | 45.8205, -121.9519 | VU-Coc | -15.4427, 167.192 |
| ZA-Kru | -25.0197, 31.4969 | ZM-Mon | -15.4378, 23.2528 |

*Author contributions.* JN and UW compiled the site coordinates and established the pipeline to obtain EO data from GEE, and unified formats. SW developed the processing steps with the input from MJ, MM, JN and NC. SB adapted the processing to Landsat data. SE provided model coefficients, code and guidance on its usage for the LST geometrical correction. SW and UW created the files that are offered to the community. TE computed flux footprints for the example sites and analysed them with respect to the satellite data together with SW and SB. SW wrote the manuscript with contributions from all authors.

*Competing interests.* The authors declare no competing interests.

*Acknowledgements.* We thank the team at the ICOS Carbon Portal for their support in publishing the FluxnetEO data sets, great thanks in particular to Ute Karstens and Zois Zogopoulos.

SW acknowledges funding from an ESA Living Planet Fellowship in the project Vad3e mecum. MJ and JN acknowledge funding from the EU H2020 projects CoCO2 (GA 958927), VERIFY (GA 776810), and E-SHAPE (GA 820852). AVP acknowledges funding from the Max Planck Society (Germany), Russian Foundation for Basic Research, Krasnoyarsk Territory and Krasnoyarsk Regional Fund of Science, project 20-45-242908. FS and CB acknowledge funds from the German Federal Ministry of Food and Agriculture (BMEL) received through the Thünen Institute of Climate-Smart Agriculture. SB acknowledges funding from the European Union through the BIOMASCAT (project code: 4000115192/18/I/NB) (https://eo4society.esa.int/projects/biomascat/) and VERIFY (project code: BO-55-101-006) (https://cordis.europa.eu/project/id/776810) projects.

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
