# Peer review of "Technical note: A view from space on global flux towers by MODIS and Landsat: The FluxnetEO dataset"

_Biogeosciences, 2021_

## Author Comment (AC2)

The manuscript by Walther et al. presents the technical details of a new customized and gap-filled remote-sensing product generated from MODIS and Landsat instruments across Fluxnet sites. They proposed a procedure to extract, qualityfilter, correct, and gapfill MODIS and Landsat data and develop standardized data products of surface reflectance, vegetation indices, and land surface temperature.

Overall, I think this is a great initiative, and I agree with the authors that "the data sets can widely facilitate the integration of activities in the ï¬·elds of eddy-covariance, remote sensing, and modeling". I also appreciate the authors' efforts in presenting the details and being frank on the merits and limitations. Therefore, I would recommend the manuscript to be considered published in Biogeosciences after addressing a few general and specific comments.

> Thank you very much, Housen Chu, for your reviewing efforts and your suggestions in support of improving the work in FluxnetEO. We answer to your questions point-by-point below in the indented blocks. Any sentences after re-phrasing we cite without line numbers as the final revised manuscript is not completed yet.

[1] I agree with the general comment made by Michael Dietze on the need to differentiate this proposed data product from the one distributed under ORNL DAAC or any other ones. Consider highlighting the uniqueness of this product or main differences with others in the manuscript. It'll help the potential users to choose a suitable product for specific use.

> We fully agree that a differentiation of FluxnetEO from complementary work is necessary to aid potential users in their decision which data set fits their application and needs best. While this is certainly not possible to do exhaustively, we decided to focus on three examples (two for Landsat, one for MODIS) plus the ORNL DAAC subsetting tool in the question from Michael Dietze. We include a paragraph in the introduction as well as a table in the supplement giving an overview of the key characteristics of the different data sets. The introductory paragraph reads like this:

*"A number of similar initiatives provide EO data that can be analysed at site-level as well. Amongst others, Robinson et al. (2017) offer 30~m Landsat NDVI for all pixels in the CONUS every 16~days between 1984-2019. The data are free from cloud effects but not necessarily from the influence of snow and water, and climatological averages over a user-defined reference period fill missing values. Moreno-Martinez et al. (2020) applied a bias-aware Kalman-filter to fuse Landsat and MODIS surface reflectance to a gap-free and smoothed product. It is mostly free of cloud, snow and water effects and contains monthly surface reflectance and their uncertainties in 6 Landsat spectral bands at 30m resolution for the CONUS. An example product for gap-free MODIS surface reflectance (as well as albedo and BRDF parameters) at approximately 1~km resolution is MCD43GF. In this case, the time series of the parameters of the bidirectional reflectance distribution function are temporally and spatially gap-filled for days and pixels with bad inversion quality or cloud and snow influence, and from those gap-free model parameters a global gap-free product of surface reflectance is provided for the MODIS land bands and three broad spectral bands. Finally, a sub-setting tool by the ORNL DAAC facilitates access to a range of global EO data sets at a large selection of sites.*
*FluxnetEO is unique in that all pre-processing steps necessary for many scientific applications at site-level are completed in its data sets: sub-setting global EO data for an area around a site, control for good quality retrievals as well as cloud, snow and water effects, and estimating missing data points in a flexible and ecologically meaningful way. Flags inform on the origin of*

*the data (i.e. original or estimated) allowing the user to make informed decisions on which data points to include the analysis and which not. While FluxnetEO is analysis-ready, it also means limited flexibility for the users to make their own decisions. For example, they depend on the site selection made by the authors (table E1) and their decision to cover an area within a radius of 2km of a site. Conversely, the ORNL DAAC subsetting tool offers larger cutout radii of 4km around a considerably larger collection of sites than FluxnetEO, and from a complementary selection of global EO products. But users will need to invest considerable work on quality control and gap-filling. Others offer quality controlled and gap-free large scale or even global EO data but the user needs to find ways to access these data sets at site level (while Moreno-Martinez et al. (2020) is available on Google Earth Engine, MCD43GF is not, Robinson et al. (2017) needs shape files), and needs to understand whether the applied quality filters match their needs."*

[2] A standardized and operationally feasible procedure for quality control and gap-filling of MODIS and Landsat data is the main focus of this manuscript. I think the authors should consider adding more analyses to validate or at least demonstrate the uncertainties/limitations of the proposed data product. The examples presented in section 4.2 are great and illustrative, but I think it needs more generalized information on the performance across sites. For example, consider comparing the data product with other available gap-filled products (e.g., MCD43GF or others like Robison et al., 2017). Also, why isn't Landsat land surface temperature included, and why do Landsat data only cover till 2017?

> We agree that a benchmarking exercise is very important for work like FluxnetEO and that we have not done it sufficiently so far. At the same time, it is challenging for a range of aspects, such as no consistency in the spatio-temporal resolutions or the land surface parameters provided, the spatial extent of the data (global or regional), or the access to the data at site-level. As you say, comparable products to consider are:

- MCD43GF: daily, 30arcsec (roughly one kilometer), gap-filling of BRDF parameters and derivation of gap-free surface reflectance and albedo from this. However, this product is not available on GEE and a mass download of global data would be necessary to subset at the sites, which is not feasible.
- Robinson et al. 2017: 16day, 30m Landsat NDVI for CONUS. Access appears difficult, at least we did not succeed to download the data for sites in the CONUS.
- In addition, there is the work by Moreno-Martinez et al. 2020: bias-aware sensor fusion Modis and Landsat to Landsat resolution, monthly, 30m, surface reflectance in different bands and their uncertainties for the CONUS. This product is available on GEE, we have downloaded it for sites in the CONUS and will include a comparison of sites across the US in the revised version of the manuscript.

Another option is to compare to automated and independent imputation algorithms, such as missforest (Stekhoven et al. 2012). We ran missforest imputation for the MODIS example, using all reflectance and LST variables together with day of the year and MODIS snow information per site. We evaluate both for the actual gaps (1), and for experiments where we introduce artificial gaps in the time series (2).

ad 1) A quick comparison for the actual gaps in the MODIS EVI and AQUA daytime LST for the example sites from the current version of the manuscript is shown below: We see overall comparable (seasonal) patterns between FluxnetEO and the missforest imputation. Missforest clearly gives more reasonable results when almost no data are available in the first two years of MODIS. This is an important potential point for improvement in a future release of FluxnetEO. Conversely, missforest is problematic in longer systematic gaps, especially snow-related ones (CA-SF1), and is computationally more expensive than FluxnetEO. Overall day-to-day variability of LST is clearly lower in the missforest imputed version than in FluxnetEO.

[Figure]

2) We furthermore perform experiments to introduce artificial gaps into the time series at originally valid and good quality data points on which we validate the performance of missforest against FluxnetEO. The literature most often adds a certain percentage of artificial gaps to the data, sometimes with a special structure, sometimes just randomly. We opt for the random variant and add certain fractions of gaps in the originally valid data points. We leave the gap-filled data in to guide the gap-filling of the artifical gaps. This procedure simulates hence random gaps, similar as they would appear due to clouds or low quality inversions, but we do not test for systematic gaps such as wet season or snow. (note that the number of sites contributing to the boxplot decreases with increasing gap-fraction as less sites have sufficient valid data to sample from).

We find that overall FluxnetEO and missforest have very comparable performance in terms of NSE, with a small but consistent tendency of FluxnetEO to outperform the missforest for the higher (more realistic!) gap-fractions. The bad performance of the missforest for the NDWI variants needs further investigation.

[Figure]

We will perform similar experiments for Landsat as well.

As for the Landsat data only covering the period until 2017, we are updating the dataset to cover a time period until 2021 for each site. In addition, we will provide gapfilled Landsat thermal band from which the FluxnetEO users could compute land surface temperature using user-defined methodologies for calculating land surface temperature.

[Figure]

[Figure]

[3] It's challenging and potentially problematic to gain generalized ideas of the spatial contexts based on those few examples (section 4.3). I understand it may not be feasible to calculate flux footprints for all sites included in this study. Still, please consider leveraging the findings from previous efforts (e.g., Göckede et al., 2008 on European sites, Chen et al., 2011, 2012 on Canadian sites, Ran et al., 2016 on Chinese sites, Chu et al., 2021 on AmeriFlux sites, and Griebel et al. 2020 on Fluxnet sites heterogeneity). Those studies analyzed many sites included in this study, providing information about the flux footprints (e.g., extents, areas) that can help justify the selection of cut-out extents and area-weighted methods.

A universally 2-km cut-out may be a bit small for specific tall tower sites. I'd suggest expanding the extents for at least the tall tower sites (e.g., forests, known tall tower (e.g., US-PFa). In our recent study (Chu et al. 2021), we found a few AmeriFlux sites (e.g., US-ChR, US-Wrc) have footprints (i.e., monthly climatology, 80% contour, based on Kljun et al., 2015 model) extending beyond 2 km from the tower. And, more sites are extending beyond 2 km if using half-hourly or daily footprints or using a different footprint model (e.g., Kormann & Meixner 2001). I think it's practically safer to start with a larger extent and then crop the images as needed.

> We agree that a universal cutout radius of 2km will not contain the flux footprint at its full extent of some sites and therefore be insufficient for detailed analyses at those sites. At the same time, 2km will suffice for a large (probably the major) part of global flux sites. There is no cutout size that fits every possible site or study goal. We therefore decide to stay with a cutout radius of 2km for the current release of the data set as a starting point, but emphasize that we do not see FluxnetEO as a static data set. Rather, we expect it to evolve and improve based on user requirements and feedback, where a larger/ more customized cutout radius is one important aspect of potential improvement in future releases.

[4] Last, I'd suggest the authors and the team consider adding other sites at this or future release, especially those with compatible processed flux datasets. For example, AmeriFlux begins rolling out processed flux data products compatible with FLUXNET2015 (see links below). Also, with other new Fluxnet initiatives (e.g., Fluxnet Co-op), it's optimistic to anticipate similar Fluxnet products will become available at more sites in the near future. As pointed out earlier, one of the major differences between this and DAAC subset products is the number of sites that are included. It will benefit many users if this data product could be generated at more Fluxnet sites.

> Good point, thanks! We compile a list of additional sites and will extent the selection of sites accordingly together with the revision of the manuscript.

https://ameriflux.lbl.gov/data/download-data-oneflux-beta/
https://ameriflux.lbl.gov/data/data-availability/#/FLUXNET/

Specific comment
[5] Line 3: Please consider adding AmeriFlux to the list as other regional networks.

> Done.

[6] Line 6-7: This sentence "…support the training and validation of ecosystem models" is vague. Consider rewriting it.

> Changed to "*support the benchmarking of ecosystem models.*"

[7] Line 95: Is the sensor difference (e.g., among Landsat 4, 5, 7, 8) or sensor drifting corrected? Also, Landsat 7 is known for Scan Line Corrector (SLC) failure and causes problematic data in certain themes. How does it be addressed?

> In order to deal with sensor difference among Landsat 4, 5, 7 and 8, we will apply the sensor harmonization approach developed by Roy et al. (2016) on the Landsat collection 2 data currently under processing to achieve consistent spectral characteristics across Landsat TM, ETM+, and OLI sensor. While this harmonzation allows us to reduce the effects of missing observations from ETM+ SLC-off gaps, gaps due to ETM+ SLC-off still remained and need to be gapfilled using the presented gapfilling approach. Note that the data collected after the SLC failure are still useful becuase they maintain the same radiometric and geometric corrections as data collected prior to the SLC failure. In the collection 1 Landsat data currently available from FluxnetEO no correction has been applied.

[8] Line 135-137: This sentence is unclear. Could you explain it a bit more in detail?

> We split this sentence and this part now reads as: *" A number of possible applications will require the analysis of actual observations, and consequently approaches that fit smooth functions to available good quality data (e.g. Jonsson and Eklundh 2002, Gonsamo et al. 2013) to represent a gap-free time series are not suitable. The idea therefore was to retain the good quality data and make as realistic estimates as possible for the gaps between them. "*

[9] Line 139-161: Does the gap-filling procedure apply to the raw bands only (i.e., calculate vegetation indices based on filled bands), or separately for both the raw bands and vegetation indices? Any justification?

> The latter one applies: the gap-filling is the last step in the processing chain and always happens after quality control, computation of a vegetation index (if not the surface reflectance is the target variable itself), spatial aggregation across subpixels (in the 'average_cutout' version). Although people will have different philosophical approaches to this question, our idea is to produce estimates in gaps of the vegetation proxy of interest itself (as opposed to e.g. the MCD43GF product which follows the approach of first gap-filling the parameters of the BRDF function and then computing the final product). Filling first the surface reflectance in spectral bands might lead to a vegetation index being computed from a mix of observed and gap-filled surface reflectances for a given data sample. We consider this problematic as of course gap-filled data points are more questionable than good quality observed ones and might not truly obey the inter-dependencies between the spectral bands, which can result in more spurious values of a vegetation index than a gap-fill estimate for the vegetation index itself in gaps where surface reflectance in one or more bands is not of sufficient quality.

[10] Line 170-172: Consider adding more granular details of the flags. Does it indicate which method is being used, or is it simply a binary flag (filled/original)?

> We clarified this in rephrasing to: *"FluxnetEO therefore provides for each data layer a gap-fill flag, that consists of a range of integer values to identify original good quality data (flag=0) from gap-filled estimates (flags=1...n) where also information is provided in which gap-filling step a certain data sample has been imputed. This allows users to explore individual sites and use (parts of) the gap-filled data or resort to only using the high quality original data points. "*

[11] 4.1 & Figure 1: Please add some discussions on the Landsat availabilities. Also, would it be more suitable to group sites by regions or biomes given geo-patterns of cloudiness? Looking at the low availability at some sites, I wonder whether it is more appropriate to leave out those sites entirely.

> At some sites data availability is indeed very low and in a second version release of FluxnetEO we will consider to remove those sites with insufficient data availability from the data sets. Please also consider that in Fig.1 we look at the central pixel of the MODIS cutout only. Data availability in spatial aggregations across subpixels will be higher in many cases. In Figure 1 our main interest was to give an idea of the overall range of gap-fraction across sites. We agree that a different grouping might reveal other apsects of the patterns, e.g. according the Koeppen climate classes. We will include a similar plot for this. We will also include a corresponding analysis for Landsat.

[12] 4.2, Figure 2-3: Please add some discussions on the Landsat time series.

> Indeed, given the very different temporal resolutions of MODIS and Landsat, an analogous presentation of Landsat is necessary. We decided for an average over a radius of 250m and will present analogous analyses for Landsat in the revised manuscript.

[13] Table 2: Please add some details about the two cut-outs to the Method section in the main text. Consider briefly justifying the weighted approach.

> We deliberately left the description of the two cut-outs out of the method section to focus here on the quality control and gap-filling. Instead, the two cut-outs are described in the data availability section. For clarity, we will also remove the reference to the two cut-outs from table 2, and add a justification of the weighted approach to the data availability section. *"The inverse distance to the tower serves as weight in the average to account for the fact that areas farther away from the stations contribute less to the measured fluxes than the immediate surroundings of a site also in the average of land surface characteristics."*

[14] Line 274-290: I suggest moving this part of the literature review to an earlier section.

> We will consider to move this part to the introduction where it fits certainly well.

[15] Line 327-340 & Figure 6: The comparison is misleading. The net radiometer (for measuring long-wave radiation) has a fixed field of view depending on its mounting height and location. It is more appropriate to compare Tsurf with LST at pixels corresponding to the radiometer's field of view or compare LSTfpa with sensible heat fluxes (or derived aerodynamic surface temperature (see Novick & Katul 2020).

> We appreciate the comment and fully agree that the net radiometer derived surface temperature cannot be expected to fully represent the temperature changes in footprints. Instead of following your suggestions we decided to go down a different road that we hope the reviewer appreciates. We followed the big leaf approach and calculated the surface temperature based on the inversion of the bulk transfer equations from the sensible heat flux using the following equation:
Tsurf = Tair + H / (ρ * cp * Gah)
with
Tair: air temperature
H: sensible heat flux
ρ: density of the air
cp: specific heat capacity of the air
Gah = 1 / (Ra_m + Rb_h): aerodynamic conductance to heat
Ra_m= u/ustar^2 : the aerodynamic resistance to momentum
Rb_h= 6.2*ustar^-(2/3): canopy boundary layer resistance for heat.

We therefore updated figure 6 with the new subplot "a". We want to highlight here that the new figure and the original one look very similar because the Tsurf calculated from the longwave outgoing radiation and the one derived from H are highly correlated as shown in subplot "b" below. We will change the respective parts of the manuscript.

[Figure]

[16] Figure 7: Consider adopting the same color scale for all EVI maps.

> We will adapt the color scale in the revised manuscript to consistent limits.

[17] Figure D1: Consider using a similar layout (e.g., extents, x-/y-axes, color codes…) as in Figures 5-6.

> Our focus in Fig.5-6 is on how the flux footprint relates to the remotely sensed surface characteristics. As the footprints on the example days chosen are relatively small, we decided to crop the Landsat cutout to a smaller extent. Figure D1 on the other hand shows the full extents of the cutouts as we are primarily interested how homogeneous or not the relationship between GPP and the EVI is across the full extent of the cutouts. This explains their different extents and being "rounded" versus rectangular. The color codes differ as well on purpose, as Fig. 5-6 show the actual EVI, while Fig.D1 shows correlations.

Chen, B., Coops, N. C., Fu, D., Margolis, H. A., Amiro, B. D., Barr, A. G., et al. (2011). Assessing eddy-covariance flux tower location bias across the Fluxnet-Canada Research Network based on remote sensing and footprint modelling. Agricultural and Forest Meteorology, 151(1), 87-100.

Chen, B., Coops, N. C., Fu, D., Margolis, H. A., Amiro, B. D., Black, T. A., et al. (2012). Characterizing spatial representativeness of flux tower eddy-covariance measurements across the Canadian Carbon Program Network using remote sensing and footprint analysis. Remote Sensing of Environment, 124, 742-755. DOI: 10.1016/j.rse.2012.06.007

Göckede, M., Foken, T., Aubinet, M., Aurela, M., Banza, J., Bernhofer, C., et al. (2008). Quality control of CarboEurope flux data – Part 1: Coupling footprint analyses with flux data quality

assessment to evaluate sites in forest ecosystems. Biogeosciences, 5(2), 433-450. doi:DOI: 10.5194/bg-5-433-2008

Griebel, A., Metzen, D., Pendall, E., Burba, G., & Metzger, S. (2020). Generating spatially robust carbon budgets from flux tower observations. Geophysical Research Letters, 47(3), e2019GL085942. doi:DOI: 10.1029/2019gl085942

Kormann, R., & Meixner, F. (2001). An analytical footprint model for non-neutral stratification. Boundary-Layer Meteorology, 99(2), 207-224. doi:DOI: 10.1023/a:1018991015119

Kljun, N., Calanca, P., Rotach, M. W., & Schmid, H. P. (2015). A simple two-dimensional parameterisation for Flux Footprint Prediction (FFP). Geosci. Model Dev., 8(11), 3695-3713. doi:DOI: 10.5194/gmd-8-3695-2015

Ran, Y., Li, X., Sun, R., Kljun, N., Zhang, L., Wang, X., et al. (2016). Spatial representativeness and uncertainty of eddy covariance carbon flux measurements for upscaling net ecosystem productivity to the grid scale. Agricultural and Forest Meteorology, 230, 114-127.

Chu, H., Luo, X., Ouyang, Z., Chan, W. S., Dengel, S., Biraud, S. C., et al. (2021). Representativeness of Eddy-Covariance flux footprints for areas surrounding AmeriFlux sites. Agricultural and Forest Meteorology, 301-302, 108350. doi:https://doi.org/10.1016/j.agrformet.2021.108350

Novick, K. A., & Katul, G. G. (2020). The Duality of Reforestation Impacts on Surface and Air Temperature. Journal of Geophysical Research: Biogeosciences, 125(4), e2019JG005543. doi:https://doi.org/10.1029/2019JG005543

Robinson, N. P., Allred, B. W., Jones, M. O., Moreno, A., Kimball, J. S., Naugle, D. E., et al. (2017). A Dynamic Landsat Derived Normalized Difference Vegetation Index (NDVI) Product for the Conterminous United States. Remote Sensing, 9(8), 863.

> Álvaro Moreno-Martínez, et al., Multispectral high resolution sensor fusion for smoothing and gap-filling in the cloud, Remote Sensing of Environment, Volume 247, 2020, https://doi.org/10.1016/j.rse.2020.111901.

> Daniel J. Stekhoven and Peter Bühlmann, MissForest—non-parametric missing value imputation for mixed-type data, Bioinformatics, Volume 28, Issue 1, 1 January 2012, Pages 112–118, https://doi.org/10.1093/bioinformatics/btr597

> D.P. Roy et al., Characterization of Landsat-7 to Landsat-8 reflective wavelength and normalized difference vegetation index continuity, Remote Sensing of Environment, Volume 185, 2016, https://doi.org/10.1016/j.rse.2015.12.024

---

## Author Comment (AC3)

In the technical note "A view from space on global flux towers by MODIS and Landsat: The FluxnetEO dataset", Walther et al. presents a standardized procedure to extract, gap-fill and quality control remote sensing observations around >300 flux sites. This contribution is critical to the reliable integration of remote sensing and eddy covariance measurements for understanding ecosystem functions and changes. I am in support of its publication, and my comments are meant to help improve the note and make it more clear to the audience.

> We appreciate your comments and your effort to help us improve clarity and thank the reviewer a lot! We answer to your questions point-by-point below in the indented blocks. Any sentences after re-phrasing we cite without line numbers as the final revised manuscript is not completed yet.

L34: As gap-fill is a key step in producing the dataset, perhaps it would be helpful to further clarify the general assumptions under these categories of methods. Other than the realistic considerations (i.e., generalizable, no need to use ancillary data) to do gap-fill only based on the remote sensing time series themselves, are there studies that suggest this method produce comparable results to complicated ones (i.e., the one that use ancillary meteo data).

> We understand that further benchmarking of FluxnetEO is absolutely necessary to understand the characteristics compared to other comparable data sets. In response to this and a similar question of reviewer 1, we will include in the revised manuscript:
> - a paragraph in the introduction outlining key characteristics of FluxnetEO in comparison to other products
> - a table in the SI
> - a comparison of FluxnetEO Landsat with a gap-filled Landsat product from Moreno-Martinez et al. 2020
> - experiments with an automated imputation by a missforest approach (Stekhoven et al. 2012) in the actual and also in artificially introduced gaps.
>
> For the sake of space we do not include the same example plots here a second time, but would like to refer to our answer to reviewer 1 question 1.

L38: "contribution" means "study"?

> Yes, "contribution to the scientific knowledge and tool sets" was meant, but we changed it to "manuscript" to enhance clarity and fluency of the reading experience.

L51: reference for "view zenith angles".

> We are afraid we do not understand what is meant here. In case a definition of a view zenith angle is wanted, it is the angle between the line of sight of a satellite instrument to the surface and the vertical line nadir above the observed point on the surface.

L136-137: I have some difficulties in understanding "The idea was to….instead of….". I feel the authors are arguing that their method is appropriate for the study though I cannot understand the second part of the sentence. "Valid data" means ancillary data or just the good quality data of the time series.

> We split this sentence and this part now reads as: " *A number of possible applications will require the analysis of actual observations, and consequently approaches that fit smooth functions to available good quality data (e.g. Jonsson and Eklundh 2002, Gonsamo et al. 2013) to represent a gap-free time series are not suitable. The idea therefore was to retain the good quality data and make as realistic estimates as possible for the gaps between them.* "

L154: it is not easy to understand the scaling method without carefully looking into some equations in ANN C. Perhaps it is helpful to insert some equations here, such as y = ax + b, where x means MSC while y is the non-gap filled time series. Then we can get a and b from the equation for each time window, and then apply a and b back to MSC for gap filled y.

> Thanks for your suggestion. We agree that equations strongly help to understand the processing in detail, and decided to add equations in appendix B. We also slightly rephrased point 4 of the conceptual description of the gap-filling procedure in the main manuscript for clarification. It now reads: *"Linearly regress the time series on its own median seasonal cycle. Compute a re-scaled median seasonal cycle with the obtained regression parameters and use it to fill longer gaps. Execute the regression and re-scaling in temporal moving windows as this guarantees more flexibility to correctly represent inter-annual variations in the time series and even partly accounts for changes in the shape of the seasonal cycle due to disturbances. It is, however, not suited to fill regularly recurring gaps at a certain time of the year, e.g. during rain seasons (Verger et al., 2013)."*

L175. Out of curiosity why do not use quality flag of MODIS here, any issue with the flag? By using statistical method only to remove the so-called outliers, are we risking removing some true extreme values?

> We had applied the MODIS quality flag (filtering for good quality in the mandatory flag, or other quality if the estimated LST error was smaller than 2K), and found that this filter barely removed data points. Restricting the second filter criterion to the LST error <1K systematically removed lower LST values within the variability range of LST, especially during summer and daytime, but obvious outliers were still kept. In order to keep data availability high and remove the obvious measurement or retrieval artifacts we opted for the outlier filter.

L210. See my comment above regarding the description of the scaling method.

> Thank you, also here we rephrased this sentence to: *"In temporal windows, find a linear scaling between one LST time series and its own MSC. Use the slope and intercept parameters to compute a re-scaled MSC which fills gaps in the time series for days of year when the MSC is valid."* "

L336. From Fig. 6a it is not accurate to say LST is consistently 30% higher, it is only the slope that is around 1.3.

> Correct, thanks a lot for this. We will also modify this analysis part in response to a suggestion from reviewer one and will correct this part accordingly in the revised manuscript.

L338. Do we really see the "slope decreases markedly for the highest temperature"? The figure only shows that slope increases a bit with the height.

> We did not intent to refer to the measurement height here with the word "high" but to the peak temperatures. We will rephrase (see also the last comment) and hope to clarify.

L389. For those sites with footprint less than 1km (which I think many sites are), how to define this aggregated snow flag. Are they either 0 or 1?

> The snow flag is 0 or 1 for each 500m subpixel in the cutout. Any aggregation across a selection of subpixels in the cutout (within 1km, across a flux footprint,etc.) will follow the procedure described by averaging the snow flag across the selected subpixels, resulting in a value between 0 and 1, or missing if less than 50% of the selected subpixels had valid information on the snow status.

L401 – 404. I am also wondering the rationale for choosing mean seasonal cycle and median seasonal cycle in different datasets. I also have to say in FLUXCOM mean seasonal cycle of remote sensing data was used but here the use of median seasonal cycle seems to be prevailing.

> This is an error and should read median seasonal cycle for Landsat as well. We figured that taking a median is advantageous over a mean to reduce the influence of undetected outliers, e.g. residual snow contamination in the reflectance-based processing.

L418. Valid snow cover < 60 days = snow does not occur at the site? I have a feeling the threshold is a bit large, e.g., a site with almost two months of valid snow cover might be considered to have no snow by this filter.

> This criterion is intended to identify sites with more or less regular snow cover to which the gap-filling step with a constant baseline value is applied. The criterion tests whether across the whole period that FluxnetEO currently covers (i.e. 21 years for MODIS and more than 30 years for Landsat), a certain number of snow days occurs. Admittedly, the thresholds are arbitrary, but based on investigation and testing. Sites with more or less regular snow cover cross this threshold easily. The MODIS snow flag occasionally (and supposedly wrongly) assigns snow for some days to weeks here and there to sites that do not typically experience snow precipitation. Most of those sites are identified with this filter and we can prevent to wrongly fill those gaps (often in the middle of the growing season!) with a constant baseline value. This benefit clearly outweighs the rare occasions that we might not fill actual snow gaps with a constant baseline value at sites that do not typically experience snow.

L450. To double check, do you mean for each time window we get a m and n?

> Yes, indeed.

L455. There is a redundant "[]" in the equation. Perhaps also would be helpful to explain the terms in the equations.

> The redundant brackets were removed and the description reformulated to make the meaning of the terms clear.

Álvaro Moreno-Martínez, et al., Multispectral high resolution sensor fusion for smoothing and gap-filling in the cloud, Remote Sensing of Environment, Volume 247, 2020, https://doi.org/10.1016/j.rse.2020.111901.

Daniel J. Stekhoven and Peter Bühlmann, MissForest—non-parametric missing value imputation for mixed-type data, Bioinformatics, Volume 28, Issue 1, 1 January 2012, Pages 112–118, https://doi.org/10.1093/bioinformatics/btr597

---

## Author Response (AR1)

Dear Editorial Team,

a point-by-point answer to the reviewers comments has been submitted on 25.2. 2022, also a community comment has been answered previously. The main changes to the manuscript in response to the reviewer comments are:

- include a discussion of comparable products and explain the uniqueness of our approach
- include different benchmarking experiments for our proposed methods
- include illustrative plots of the method also for Landsat and not only for MODIS